# Forecasting estuarine salt intrusion in the Rhine-Meuse delta using an LSTM model

Bas J.M. Wullems[1,2], Claudia C. Brauer[1], Fedor Baart[2,3], and Albrecht H. Weerts[1,2]

[1]Hydrology and Quantitative Water Management Group, Wageningen University, The Netherlands
[2]Department of Operational Water Management & Early Warning, Unit of Inland Water Systems, Deltares, Delft, The Netherlands
[3]Department of Hydraulic Engineering, Faculty of Civil Engineering and Geosciences, Delft University of Technology, The Netherlands

**Correspondence:** Bas Wullems (bas.wullems@wur.nl)

**Abstract.** Estuarine salt intrusion causes problems with freshwater availability in many deltas. Water managers require timely and accurate forecasts to be able to mitigate and adapt to salt intrusion. Data-driven models derived with machine learning are ideally suited for this, as they can mimic complex non-linear systems and are computationally efficient. We set up a Long Short Term Memory (LSTM) model to forecast salt intrusion in the Rhine-Meuse delta, the Netherlands. Inputs for this model are chloride concentrations, water levels, discharges and wind speed, measured at 9 locations. It forecasts daily minimum, mean and maximum chloride concentrations up to 7 days ahead at Krimpen aan den IJssel, an important location for freshwater provision. The model forecasts baseline concentrations and peak timing well, but peak height is underestimated, a problem that becomes worse with increasing lead time. Between lead times of 1 and 7 days, forecast precision declines from 0.9 to 0.7 and forecast recall declines from 0.7 to 0.5 on average. Given these results, we aim to extend the model to other locations in the delta. We expect that a similar setup can work in other deltas, especially those with a similar or simpler channel network.

## 1 Introduction

Salt intrusion occurs in estuaries around the world (Apel et al., 2020; Augustijn et al., 2011; Qiu and Wan, 2013; Rohmer and Brisset, 2017; Shaha et al., 2013; Xue et al., 2009). In an estuary, high-density seawater protrudes underneath fresh water from the river. Daily tidal motions, wind-driven dispersion and variations in coastal swell and river discharge change the position and shape of the salt-fresh interface (Savenije, 2012). During periods of prolonged drought and storms, salt water intrudes further inland than under ordinary conditions. This can hamper freshwater availability, especially in densely populated deltas such as the Hudson (Lerczak et al., 2009) Rhine (Van den Brink et al., 2019) and Changjiang (Xue et al., 2009).

In some areas, salt intrusion has been causing problems for years. In the Changjiang delta, a salt intrusion event in 1999 caused drinking water abstraction to be paused for 25 days (Xue et al., 2009). Other deltas may also be prone to such problems, as rising sea levels due to climate change are expected to increase salt intrusion and put a strain on fresh water supply, especially in areas that will experience drier summers and heavier storms (Van den Brink et al., 2019; Huismans et al., 2019; Beijk et al., 2017). While storms only last hours, they can cause elevated chloride concentrations for weeks (Huismans et al., 2018). As

a recent example, in the summer of 2022, a prolonged drought hit Europe. As a result, the discharge of the river Rhine was severely reduced for months and chloride concentrations in the tidally influenced part of the river (near Lekhaven, see Fig. 1)

exceeded 8000 mg l$^{-1}$ (Rijkswaterstaat, 2022). The yearly average of tidal maximum chloride concentration over 2022 was above 3500 mg l$^{-1}$, which occurs on average once in 24 years based on data of the 20$^{th}$ century (Beersma et al., 2005).

In some intensively managed delta areas, surface water is transferred from the larger (sometimes tidally influenced) rivers to smaller waterways through inlets, so it can be used to ensure suitable groundwater levels, flow velocity and water quality for the local land use (Brauer, 2014). This gives water managers tools to limit the consequences of salt intrusion for fresh water

availability (Prinsen and Becker, 2011). Inlets from the larger waterways to smaller channels can be closed to prevent the salt water from reaching agricultural areas. Alternatively, fresh water can be diverted from areas with a surplus to areas with salt intrusion. There, it can either be used to supplement the freshwater intake, or to push the fresh-salt interface back seawards (Augustijn et al., 2011). These decisions are usually made based on observations and operational rules (Pezij et al., 2019).

While operational rules are suitable for mitigation of freshwater availability problems on a short timescale, some of the

larger-scale measures take several days to implement. To use these mitigation tools in a timely fashion, it would be useful to have a multi-day forecast of chloride concentrations at some critical locations (Hauswirth et al., 2021). This would give water managers more time to implement measures. A physical or conceptual model can be used for that, but one-dimensional hydraulic models struggle to represent the three-dimensional nature of the salt intrusion processes, while three-dimensional models are too computationally demanding to run on operational timescales (Warmink et al., 2011; Buschman, 2018; Huismans

et al., 2016). Generalized conceptual models can capture some of the estuarine dynamics and are especially valuable when data availability is limited, but are difficult to apply in multi-branched estuaries (Savenije, 1986; Gisen et al., 2015; Sun et al., 2020).

A data-driven model, derived using machine learning, might be used as an alternative approach to this forecasting problem. Once trained, data-driven models have been reported to be successful in capturing non-linear systems (Kratzert et al., 2018), and have a runtime of milliseconds to seconds per timestep once trained (Haasnoot et al., 2014; Hauswirth et al., 2021; Zounemat-

Kermani et al., 2020). Machine learning approaches have successfully been applied to describe hydrological extremes (e.g. Hauswirth et al., 2021), shoreline evolution (e.g. Calkoen et al., 2021) and rainfall-runoff processes (e.g. Kratzert et al., 2018). There have also been some successes in salt intrusion forecasting (Hu et al., 2019; Rohmer and Brisset, 2017; Zhou et al., 2020), but the complexity of the multi-branched and strongly managed Rhine-Meuse delta has proven difficult to model with this approach, at least on hourly timescales (Korving and Visser, 2021).

The aim of this study is to test the possibilities of data-driven modelling of chloride concentrations in the Rhine-Meuse delta. As a starting point for such a data-driven model, we create a model to forecast chloride concentrations at one location on a daily basis. This model should be able to predict the occurrence of salt intrusion peaks several days to a week in advance. From there, further improvements can be made by extending the model to other locations and making it suitable for higher temporal resolutions. If we are able to create useful forecasts for this delta, a similar approach could be applied to other complex deltas.

Furthermore, if this approach is successful for a multi-branched and intensively managed estuary, we expect it to work for a single-channel, more natural estuary as well.

In this paper we present a method to forecast salt intrusion in the surface waters of the Rhine-Meuse delta using a machine learning approach. We will (1) identify a location for which a forecast would be especially helpful, (2) select the observations required to make the prediction and (3) design a suitable model architecture. We will then (4) optimize the model using suitable criteria and (5) test the model on a separate dataset. Finally, we will (6) assess the importance of each input observation for the predicted output and relate this to estuarine processes.

## 2    Material and methods

We designed a machine learning model to forecast chloride concentrations near a critical junction in the Rhine-Meuse delta. We started by exploring the study area and identifying a location for which a salt intrusion forecast is needed (Sect. 2.1). We retrieved observations of possibly relevant variables and did an exploratory analysis (Sect. 2.2). We set up a machine learning model to predict concentrations one day ahead, and optimized it using suitable performance metrics (Sect. 2.3). We then ran it to predict concentrations up to seven days ahead and used a separate dataset for testing. Finally, we performed a sensitivity analysis (Sect. 2.4).

### 2.1    Study area

The Rhine-Meuse delta is located in the Netherlands and comprises roughly half the country (Fig. 1). Near the cities of Arnhem and Nijmegen, the river Rhine splits into three branches: the IJssel, Waal and Nederrijn/Lek. While the IJssel flows north and discharges into the IJsselmeer, the Lek and Waal flow west and flow into the Hollandsch Diep, Haringvliet and Nieuwe Waterweg, around the cities of Dordrecht and Rotterdam. The Meuse enters the country near Maastricht and flows parallel to the Waal before discharging into the Hollandsch Diep. In the eastern part of the country, weirs are often used to regulate water levels and discharges. This includes some large weirs in the Nederrijn/Lek, at Driel (near Arnhem) and Hagestein. In the lower-lying, flatter western part of the country, the larger waterways cannot be managed in such a way. Water levels in the smaller channels and ditches between fields are intensively managed with weirs and are supplied with river water through inlets.

The Nieuwe Waterweg forms an open connection of the river system to the North Sea. While many other estuaries in the delta have been (partially) closed off, this waterway was kept open to ensure easy access for ships to the port of Rotterdam. It connects to the lower reaches of the Rhine-Meuse system, called the Nieuwe Maas and Oude Maas, in which the tide causes daily variations in chloride concentrations. Occasionally, the salt water intrudes further upstream and reaches the Hollandsche IJssel, a small branch within the delta that is important for freshwater provision to agricultural channels and drinking water companies in the west of the country (Prinsen and Becker, 2011; Van den Brink et al., 2019). In order to keep this branch fresh, water managers can divert water from the river Waal or from the IJsselmeer towards this area (Haasnoot et al., 2014; Prinsen and Becker, 2011). However, to do this effectively, they require predictions of salt concentrations several days ahead. A timely forecast would provide them with support for decision-making in a complex area with many stakeholders. We selected

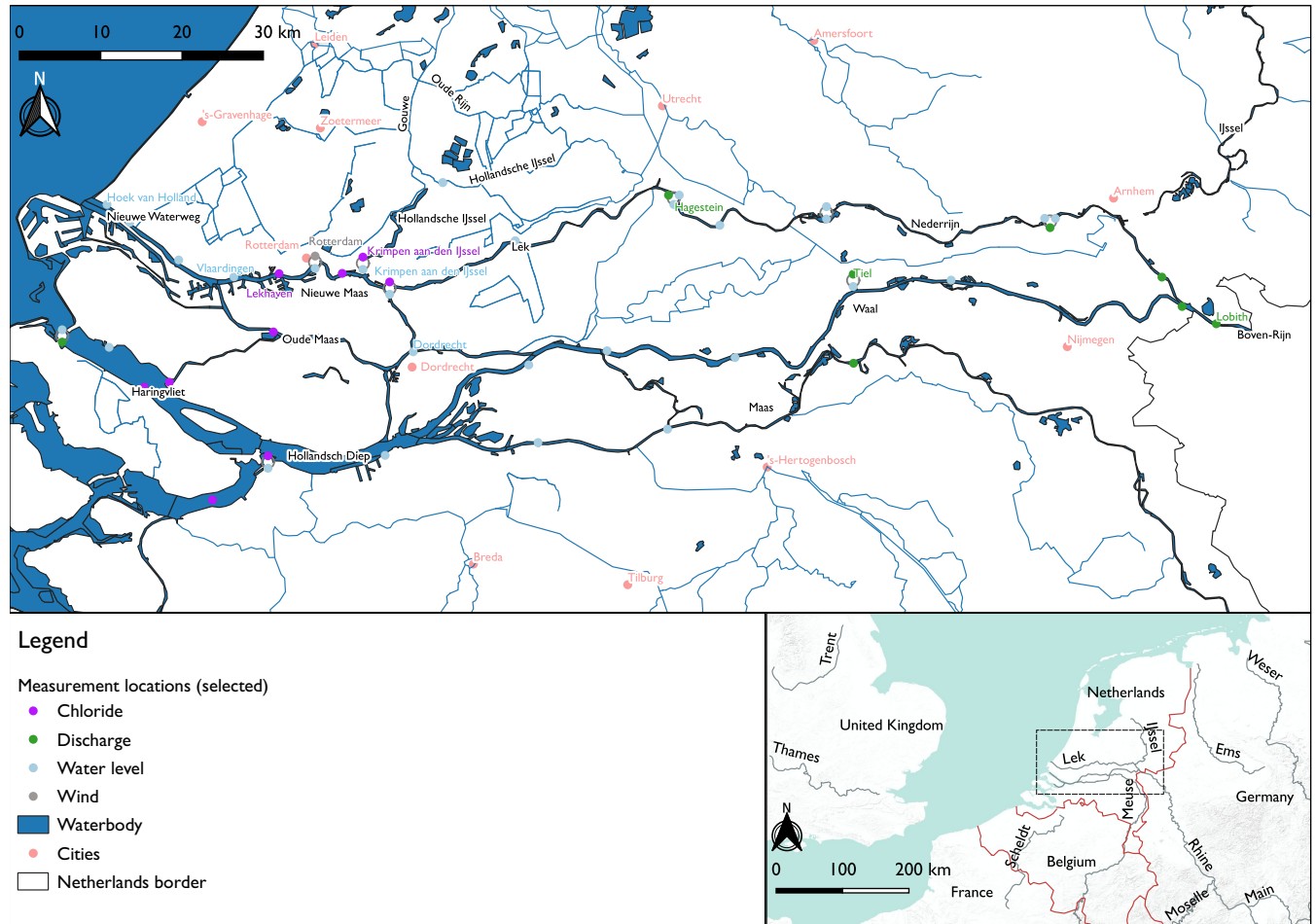

**Figure 1.** Map of the study area, indicating measurement locations that were considered for this study. Only measurement locations that were used in the model are labeled. Map created with QGIS (2022) using data from PDOK (2022), Rijkswaterstaat (2022) and KNMI (2022).

Krimpen aan den IJssel for this forecast, as it is located at the junction of the Nieuwe Maas and Hollandsche IJssel branches and has a sufficiently long record of measurements.

## 2.2  Data

### 2.2.1  Data collection

We obtained data for this research from Rijkswaterstaat (2022) and KNMI (2022). These data are published daily, which makes them suitable for operational forecasting. We selected the following variables from the years 2011–2020: discharge ($m^3\ s^{-1}$), water level (cm above mean sea level), wind speed ($m\ s^{-1}$) and chloride concentration ($mg\ l^{-1}$). Chloride is measured at one, two or three depths, depending on the location. We used discharge observations from Lobith and Tiel, for which stage-discharge

relationships are avaialable, and Hagestein, where a weir is present. For the western part of the study area, we obtained water levels in the large waterways as a proxy for discharges and pressure differences between branches. For the wind speed, we used measurements at Rotterdam, in the middle of our study area. The daily mean wind speed was decomposed into an east-west and a north-south component, using the wind direction. We used the years 2011–2017 to train the model and the years 2018–2020 to test its performance.

### 2.2.2 Timeseries exploration

We explored a large number of timeseries. This section summarizes the main findings of that exploration, with some examples shown in Fig. 2.

Salt intrusion events are quite rare. In the total 10-year period considered, there have been 127 days where chloride concentrations at Krimpen aan den IJssel exceeded 300 mg $l^{-1}$, which is twice the drinking water limit. Of these 127 days, 75 occurred in 2018.

Chloride peaks propagate upstream. Steady rises in chloride concentration at downstream locations sometimes precede upstream rising concentrations (e.g. Fig. 2(a,b), Sep 2017). However, downstream rising concentrations most often coincide with only minor concentration increases upstream (e.g. Fig. 2(a,b), May 2017, Jun 2017). Instead, the biggest peaks show very pronounced spikes that are much steeper than the steady background concentration increase. These spikes coincide with increased water levels during periods of relatively high wind speed (e.g. Fig. 2(c, e), Jan 2017).

The water levels in the Nieuwe Waterweg and Nieuwe Maas branches are strongly linearly correlated. Water levels at Krimpen aan den IJssel are correlated with those at Hoek van Holland with a Pearson coefficient of 0.72. For points between these two locations, correlations are between 0.76 and 1. Water levels at Dordrecht, which is located on the Oude Maas, deviate more from the other locations. A more complete overview of water level correlations can be found in Appendix A.

Some of the larger peaks in chloride concentration coincided with high water levels at Hoek van Holland (e.g. Fig. 2(a,c), Jan 2017). This could be caused by a storm surge, possibly coupled to a spring tide. However, there are also many examples where water levels at Hoek van Holland and wind speeds at Rotterdam are high, but no increase in chloride is observed (e.g. Fig. 2(a,c,e), Nov 2017, Dec 2017).

Salt intrusion events are often coupled to low river discharges (Fig. 2(a,d), Jan 2017), but this is not always the case.

### 2.3 Model design

Figure 3 shows how we designed the LSTM model. Preprocessing steps are explained in Sect. 2.3.1. After preprocessing, we created the machine learning model architecture (Sect. 2.3.3) and assessed its performance on the training dataset using the metrics described in Sect. 2.3.4. We then adapted the model hyperparameters in steps until an optimum had been reached. The final model was then used on the test dataset (Sect. 2.3.5). Finally, we performed a sensitivity analysis (Sect. 2.4).

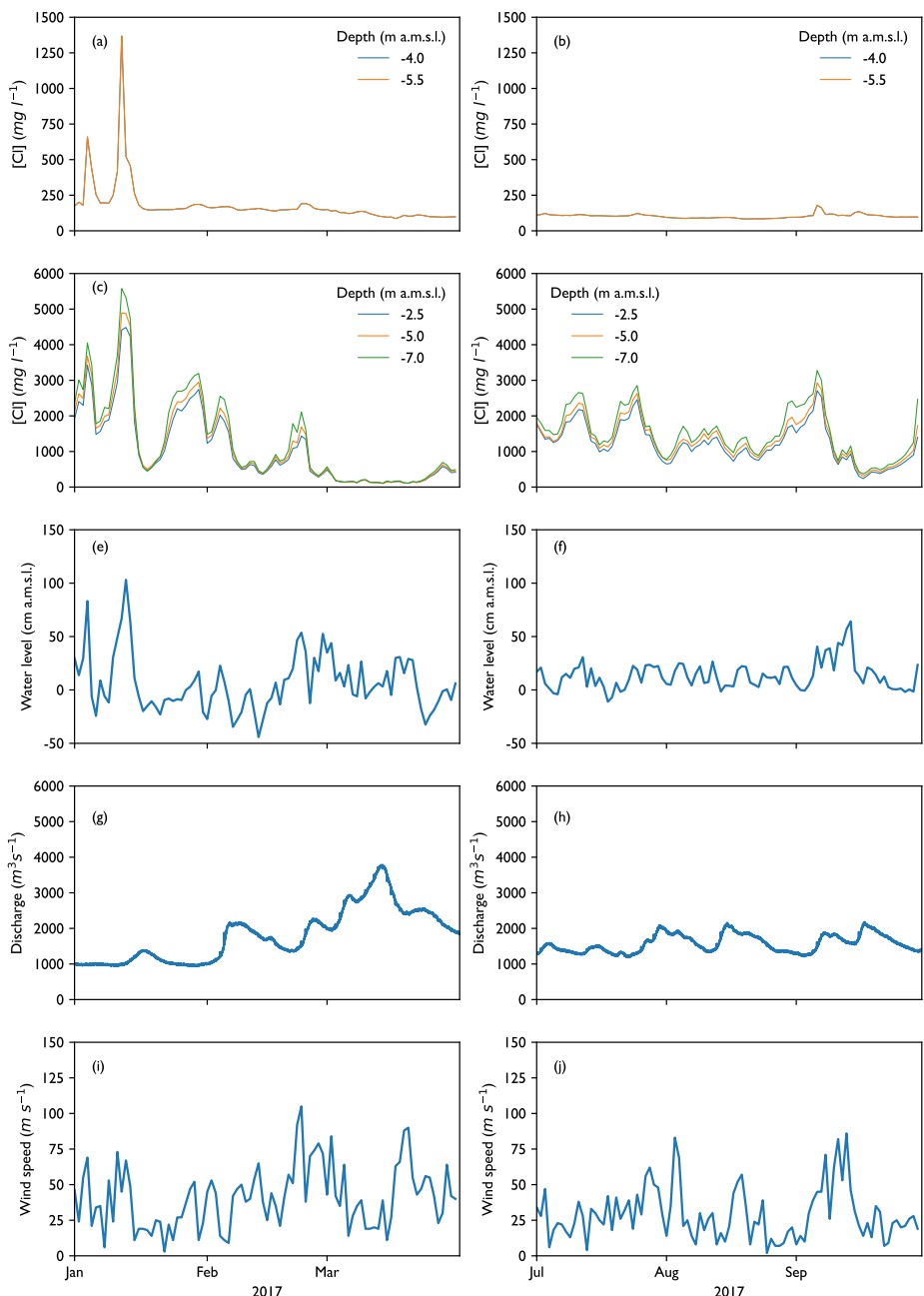

**Figure 2.** Example timeseries for January-March (a,c,e,g,i) and July-September (b,d,f,h,j) of the year 2017. Daily mean chloride concentrations at Krimpen aan den IJssel (a,b) and Lekhaven (b,d) are shown. Note that the peaks in chloride at Lekhaven in January are reflected at Krimpen aan den IJssel, while this effect is absent or much weaker in the other months. In this section, we relate this to changes in daily mean water level at Hoek van Holland (e,f), daily mean discharge at Lobith (g,h) and daily mean wind speed at Rotterdam (i,j).

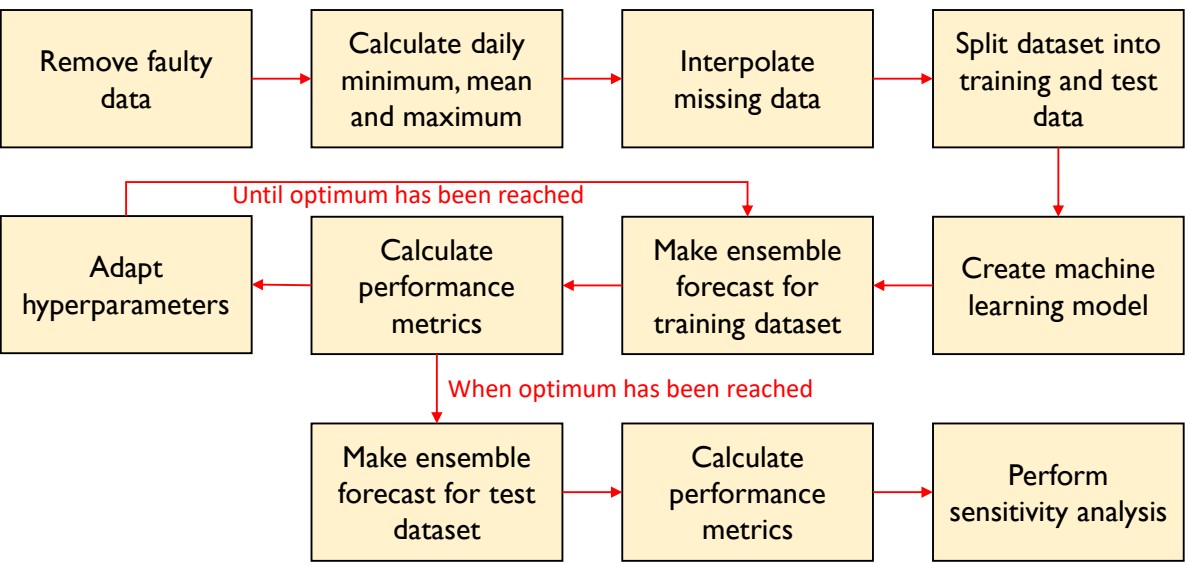

**Figure 3.** Workflow for design and analysis of the LSTM model. Preprocessing steps are shown in the top row. After preprocessing, we set up the machine learning model architecture and trained it on the training dataset. We then created an ensemble forecast with the trained model and calculated its performance on the training dataset. We calculated performance metrics and adapted model hyperparameters step by step to find an optimum. The final model was then applied to the test dataset. Finally, we perfromed a sensitivity analysis.

### 2.3.1 Preprocessing

We tried to keep data preprocessing to a minimum, to reduce computation time and make it easy to use new data. We removed extreme outliers and physically impossible values from the raw data. For every day with measurements, we then calculated daily minimum, mean and maximum for each of the variables. We then used linear interpolation to fill in any gaps in the daily features. Finally, we split the dataset into training data (2011–2017) and test data (2018–2020).

### 2.3.2 Feature selection

We selected a subset of the available features to set up the machine learning model. Reducing the number of features in a machine learning model helps to speed up its training and prevent overfitting. A first selection of features was made based on

**Table 1.** Overview of features used for chloride prediction at Krimpen aan den IJssel. A checkmark (✓) indicates whether a feature has been retained after Boruta analysis.

| Variable | Location | Statistic | Feature Name | $t_{-7}...t_{-5}$ | $t_{-4}...t_0$ | $t_{-4}...t_{+1}$ |
|---|---|---|---|---|---|---|
| Chloride (mg l$^{-1}$) | Krimpen aan den IJssel -4.0 m | min | ClKr400Min | | ✓ | |
| | | mean | ClKr400Mean | | ✓ | |
| | | max | ClKr400Max | | ✓ | |
| | Krimpen aan den IJssel -5.5 m | min | ClKr550Min | | ✓ | |
| | | mean | ClKr550Mean | | ✓ | |
| | | max | ClKr550Max | | ✓ | |
| | Lekhaven -2.5 m | min | ClLkh250Min | | ✓ | |
| | | mean | ClLkh250Mean | | ✓ | |
| | | max | ClLkh250Max | | ✓ | |
| | Lekhaven -5.0 m | min | ClLkh500Min | | | |
| | | mean | ClLkh500Mean | | | |
| | | max | ClLkh500Max | | | |
| | Lekhaven -7.0 m | min | ClLkh700Min | | ✓ | |
| | | mean | ClLkh700Mean | | ✓ | |
| | | max | ClLkh700Max | | ✓ | |
| Water level (cm a.m.s.l.) | Krimpen aan den IJssel | min | HKrMin | | | ✓ |
| | | mean | HKrMean | | | ✓ |
| | | max | HKrMax | | | ✓ |
| | Hoek van Holland | min | HHvhMin | | | |
| | | mean | HHvhMean | | | ✓ |
| | | max | HHvhMax | | | |
| | Dordrecht | min | HDrdMin | | | |
| | | mean | HDrdMean | | | ✓ |
| | | max | HDrdMax | | | |
| | Vlaardingen | min | HVlaMin | | | |
| | | mean | HVlaMean | | | ✓ |
| | | max | HVlaMax | | | |
| Discharge (m$^3$ s$^{-1}$) | Lobith | mean | QLobMean | | | ✓ |
| | Hagestein | mean | QHagMean | | | ✓ |
| | Tiel | mean | QTielMean | | | ✓ |
| Wind speed (east-west) (m s$^{-1}$) | Rotterdam | mean | WindEW | | | ✓ |
| Wind speed (north-south) (m s$^{-1}$) | Rotterdam | mean | WindNS | | | ✓ |

the observations in Sect. 2.2.2. A second selection was made with a feature selection algorithm. The full set of features and the subset used for model building are listed in Table 1.

Since a number of cases showed increasing trends in chloride concentration over a week-long period, we used chloride observations up to 7 days back to predict concentrations on a given day. For Krimpen aan den IJssel, this concerns measurements at the location itself and the downstream location of Lekhaven (see also Fig. 1 for locations). All measurements depths (two for Krimpen aan den IJssel; three for Lekhaven) were retained in this part of the selection procedure. The same 7-day window was used for the other variables. The strong correlation between water levels at different locations suggests that it is safe to exclude most stations without losing unique information. Therefore, we used water levels from four locations: Krimpen aan den IJssel, the two downstream locations Hoek van Holland and Vlaardingen, and Dordrecht, to account for pressure differences between the northern and southern parts of the estuary which drive flow between the branches. Discharges from three upstream locations are included: Lobith, where the Rhine enters the Netherlands and for which forecasts are derived; Tiel, representative for the Waal branch; Hagestein, representative for the Nederrijn/Lek branch. We used observations of wind speed at a single station, Rotterdam, which is located in the middle of our study area. For chloride and water level, daily minima, means and maxima are included, to account for the rapid subdaily fluctuation. For discharge, we only use the daily mean, as subdaily fluctuations are small.

We performed a second feature selection using the Boruta algorithm (Kursa and Rudnicki, 2010; Homola et al., 2022). With this algorithm, a linear regression model is fitted using decision trees. The fitting process consists of several iterations. At each iteration some of the features are replaced by shadow features, which are randomized copies of the original features, effectively replacing information for that feature by noise. The algorithm then tests if removing this information made the model perform any worse. A feature is supposed to be more important when the prediction quality deteriorates more when that feature is replaced. This way, the features are ranked by relevance. We did this three times, with daily minimum, mean and maximum chloride concentration at Krimpen aan den IJssel at a depth of -4.00 m a.m.s.l. as target variables. Results of the Boruta analysis showed that for most variables, four or five timesteps are relevant for prediction of the output variables. Which timesteps these are exactly varies. We decided to retain only timesteps ranging from $t_{-4}$ to $t_{+1}$. The timestep $t_{+1}$ is only used for discharge, water level and wind speed. This mimics a situation where these variables have already been forecast using another model which does not include salt intrusion forecasting. In addition, some variables were omitted altogether, since they do not provide information that the retained variables do not already provide. Finally, we only used daily means of water levels for all locations except Krimpen aan den IJssel. A more detailed motivation for the choices we made can be found in Appendix B. The final selection of variables is given in Table 1.

### 2.3.3 Model architecture

We set up a Long Short Term Memory model (LSTM) to predict chloride concentrations up to 7 days ahead using the variables in Table 1. An LSTM is a specific type of neural network model designed by Hochreiter and Schmidhuber (1997). While in an ordinary neural network model variables are being fed into nodes and given weights, an LSTM cell takes a sequence as input and can learn not only the weight to be given to such a sequence, but also the timesteps which are useful to remember for the

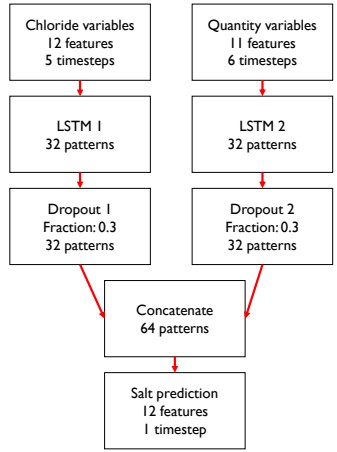

**Figure 4.** Schematic overview of the machine learning model. Quantity variables are water level, discharge and wind speed.

prediction of a new value. This makes LSTMs especially suitable for applications with a sequential nature, such as language processing and timeseries analysis. Indeed, in an exploratory analysis we found that the LSTM model's predictions were closer
to observed values than those of a feedforward neural network or a multivariate linear regression model, using the same input features.

      We set up the model using the tensorflow and keras packages in python (for documentation, see Abadi et al., 2015; Chollet, 2015). We used scikit-learn (Pedregosa et al., 2011) for preprocessing. Measurements of chloride concentration for $t_{-4}$ up to $t_0$, as well as measurements of discharge, water level and wind speed for $t_{-4}$ up to $t_{+1}$ were used as input, as indicated in
Table 1. For each variable in the training dataset, we calculated mean $\mu$ and standard deviation $\sigma$ and then converted each value $x$ to its normalized value $z$ using

$$z = \frac{x - \mu}{\sigma} \quad . \tag{1}$$

The same scaling, with $\mu$ and $\sigma$ derived from the training dataset, was applied to the test dataset.

      The structure of the LSTM model is shown in Fig. 4. Because the chloride input timeseries are five steps long and the water
level, discharge and wind timeseries are six steps long, we split the data into two groups. The first group contains all chloride concentrations and the second group contains the other three variables (i.e. water level, discharge and wind speed), that are hereafter also referred to as 'quantity variables'. The chloride timeseries and the quantity timeseries are fed to separate LSTM layers, which are used to recognize developments in the variables over time. The LSTM layer contains many parameters, such as the weight given to each input feature and the timesteps for which this feature must be retained. Each of these parameters is
optimized in the machine learning algorithm (Sect. 2.3.5). The outputs of these LSTM layers are then concatenated and fed into a dense layer, which applies weights to these intermediate outputs to end up with the chloride concentration at Krimpen aan

den IJssel at $t_{+1}$. As a protection against overfitting, a dropout layer is added between the LSTM layers and the concatenation layer. A dropout layer randomly sets some inputs to zero at each iteration of the training procedure, thereby making it less likely for the model to obtain a perfect fit for the training dataset and forcing it to account for some noise. This makes the model more likely to perform well in a new situation.

The model is trained to predict chloride concentrations on $t_{+1}$, which is the first forecast. The forecast is then added to the record of chloride concentrations and used to forecast the next timestep. The length of timeseries used to make a forecast remains the same, so to make a forecast for $t_{+2}$, chloride concentrations from $t_{-3}$ to $t_{+1}$ are used. This procedure is repeated to forecast chloride concentrations up to $t_{+7}$. We chose this approach rather than training separate models for each lead time, as the latter approach might lead to feature weights suddenly shifting from one timestep to the next, which makes the results hard to interpret.

### 2.3.4 Performance metrics

The Root Mean Square Error (RMSE) is a measure for the deviations between the predicted and observed value of a variable. It is calculated as

$$\text{RMSE} = \sqrt{\frac{1}{n}\sum_{i=1}^{n}(\hat{y}_i - y_i)^2} \tag{2}$$

in which $y_i$ is the i[th] observation of the target variable, $\hat{y}_i$ is the model estimate of the target variable and $n$ is the number of observations.

Forecast quality can be expressed in the metrics precision and recall. For this, an event threshold is defined at a daily mean chloride concentration of 300 mg l[-1], which is twice the drinking water limit (Van den Brink et al., 2019), as an indicator for severe salt intrusion. When the model predicts [Cl] above 300 mg l[-1] for a certain day, this is defined as a warning. Each day on which the observed value for [Cl] exceeds 300 mg l[-1] is defined as an event. Consecutive days with chloride concentrations above the threshold are considered multiple events. Precision and recall can then be calculated as

$$\text{Precision} = \frac{|\text{Events} \cap \text{Warnings}|}{|\text{Warnings}|} \tag{3}$$

and

$$\text{Recall} = \frac{|\text{Events} \cap \text{Warnings}|}{|\text{Events}|} \tag{4}$$

where |Events| indicates the number of events, |Warnings| the number of warnings and |Events ∩ Warnings | the number of events for which a warning was issued. A high precision indicates that the warnings issued by a model are often justified. High recall indicates that events are often captured by the model.

**Table 2.** Tuned hyperparameters for the LSTM model

| | |
|---|---|
| Size LSTM 1 | 32 |
| Size LSTM 2 | 32 |
| Batch size | 64 |
| Extra hidden layer size | 0 |
| Dropout (after LSTM) | 0.3 |
| Dropout (after extra hidden layer) | N/A |
| Weights of output variables: | |
| -ClKr400Min | 2 |
| -ClKr400Mean | 3 |
| -ClKr400Max | 3 |
| -ClKr550Min | 1 |
| -ClKr550Mean | 1 |
| -ClKr550Max | 1 |
| -ClLkh250Min | 1 |
| -ClLkh250Mean | 1 |
| -ClLkh250Max | 1 |
| -ClLkh700Min | 1 |
| -ClLkh700Mean | 1 |
| -ClLkh700Max | 1 |

The performance of the LSTM in terms of these metrics is compared to a persistence forecast, which functions as a baseline. The assumption of a persistence forecast is that future chloride concentrations are the same as on the current day, i.e. $[Cl]_{t_0} = [Cl]_{t_{+1}} = [Cl]_{t_{+7}}$.

### 2.3.5 Model tuning and testing

We further optimized the general model architecture described in Sect. 2.3.3 by tuning several hyperparameters (Table 2). The sizes of both LSTM layers were adjusted in steps and model performance in terms of RMSE, precision and recall was recorded. The same was done for the presence and size of an extra hidden layer, and for the dropout parameter. Finally, weights were given to the twelve output variables of the model. When a variable's weight is larger, the learning algorithm penalizes errors in the prediction of that variable more than that of other variables.

For each set of hyperparameters, we trained three models. Each model starts with different initial parameter weights, which are random. These weights are then applied to the input variables to calculate the output variables. The quality of the model is calculated as a mean squared error. The parameter weights are then adjusted and the calculations are redone. For this adjustment, we used the adam optimizer (Kingma and Ba, 2017), which is able to determine the optimal size of an adjustment

step. The models were trained to predict chloride concentrations one day ahead. We then used them to create a forecast up to 7 days ahead, as described in Sect. 2.3.3. RMSE, precision and recall at $t_{+1}$, $t_{+4}$ and $t_{+7}$ were recorded for each model training run, yielding 9 values for each metric per set of hyperparameters. By comparing these metrics, we determined the optimal values for the hyperparameters (Table 2). A full overview of tuned hyperparameters and metrics can be found in Appendix C.

The hyperparameter setup in Table 2 was then re-used for training an ensemble of 15 models, as the ensemble mean RMSE was shown not to change markedly anymore when ensemble size was increased further. The ensemble is created by fitting the model multiple times, with slightly different initial parameter weights each time. Each model from the ensemble was then used to forecast chloride concentrations in the testing period.

## 2.4   Sensitivity analysis

We performed a sensitivity analysis on the ensemble of models to investigate how variations in the input variables impact the predicted value of mean daily chloride concentration. To do this, we perturbed each input variable in the test dataset by adding 0.2 (in normalized units) to all values of that variable, while keeping the values of the other variables the same. This way every variable is increased by an amount that is within its normal range, but markedly higher than the normal situation. Choosing a higher value for the deviation might show clearer dependencies, but would be a further departure from what is physically realistic. It might also lead to impossibilities such as negative chloride concentrations. The model ensemble was then rerun for each perturbed variable separately. We then calculated the average difference in chloride concentration between the original dataset and the perturbed dataset. This gives an indication of the sensitivity of mean daily chloride concentration to changes in each of the other variables. We want to stress that many of these changes are not physically realistic, as most variables we consider would not change independently of the others. However, it gives an indication of the weight the model gives to each variable. In a linear regression model, we would simply use weight parameters to show this, but the complex structure of the LSTM model makes weights difficult to interpret. Therefore, we chose this method to show a general relation between model input and output.

## 3   Results

### 3.1   Model performance on training dataset

Figures 5(a,c,e) show the forecasts made for the year 2011, during which a number of high chloride concentration peaks occurred in autumn. This includes the highest value in the training dataset, measured on 28 November 2011. The forecast values follow the observations closely and continue to do so for lead times over 3 days, which indicates that the model architecture is complex enough to capture complexities in the dataset. The largest peaks, however, are still often underestimated. Predictions of the full training dataset (2011–2017) match the observations well and have no systematic bias (Fig. 5(b,d,f)). When we define an event as a day with chloride concentrations >300 mg l$^{-1}$, we see that these are generally well captured for the training

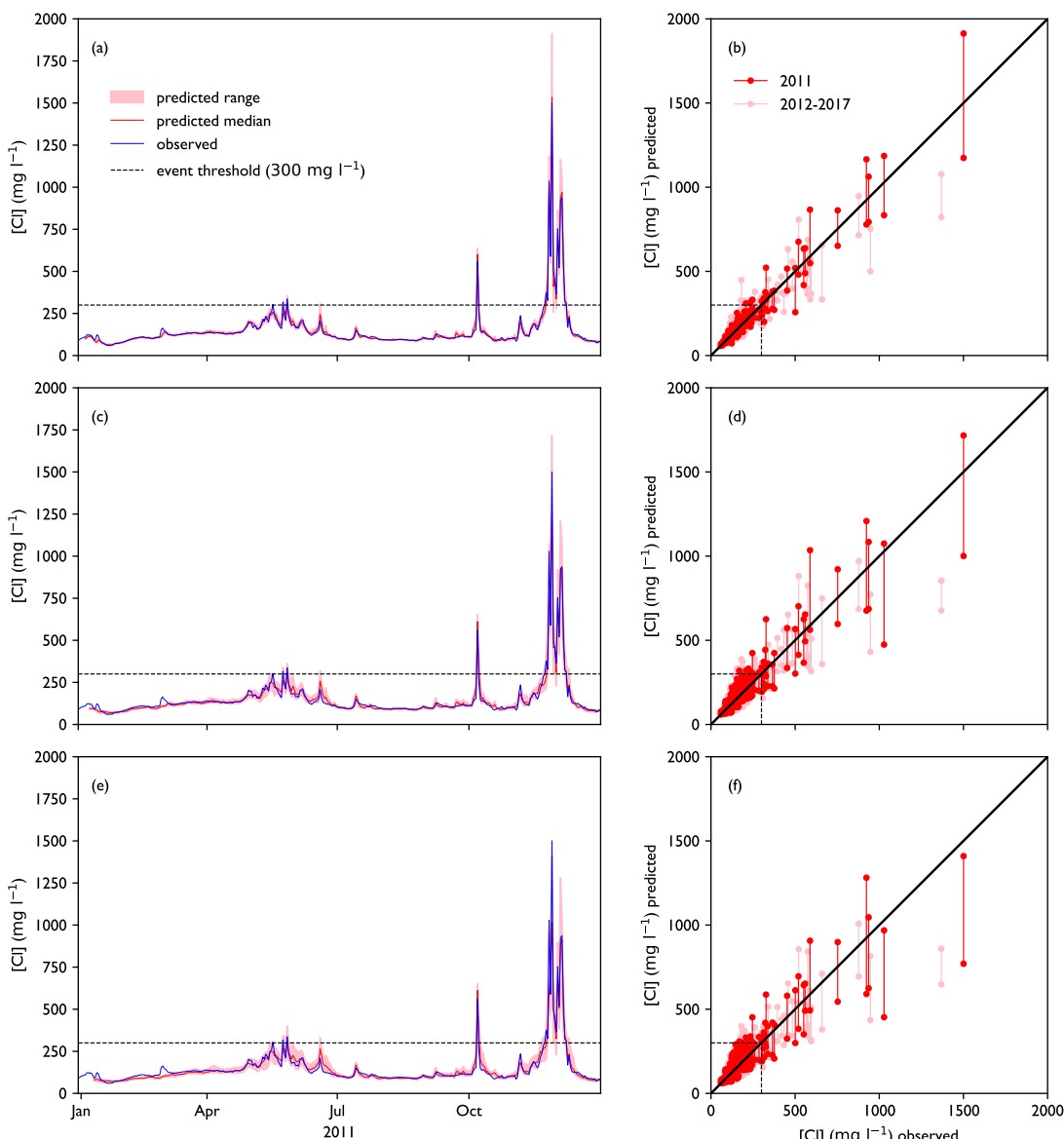

**Figure 5.** Model performance for the training period (2011–2017). Panels (a), (c) and (e) show a collection of forecasts of ClKr400Mean made with lead times of (a) 1 day, (c) 4 days and (e) 7 days for the year 2011, along with observed values. The predicted value is given as an ensemble prediction for each day of the year, created with the lead time mentioned. Median and range of the ensemble prediction are shown. Panels (b), (d) and (f) show predicted vs. observed values for the full training dataset with lead times of 1, 4 and 7 days, respectively. For each day in the training dataset, a vertical line indicates the range of predicted values by the ensemble members. A figure showing all ensemble members separately can be found in Fig. D1.

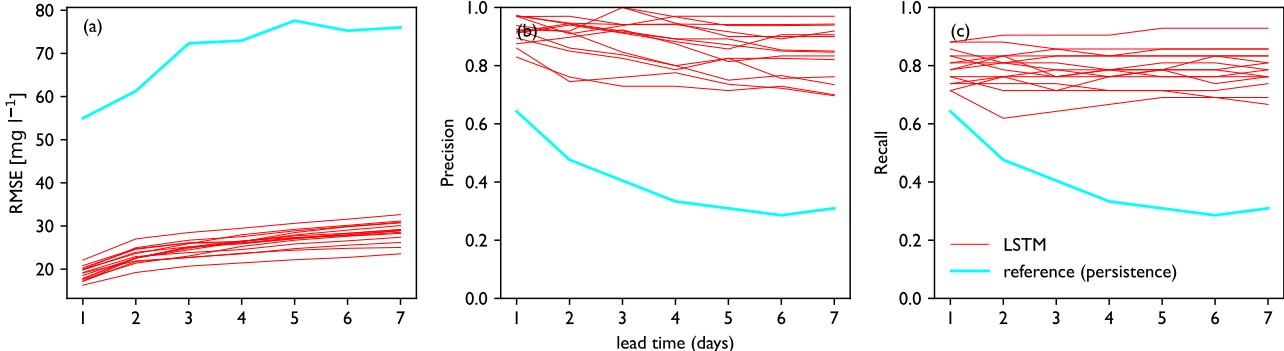

**Figure 6.** Performance metrics vs. lead time for the training period. Panels show (a) RMSE, (b) precision and (c) recall. Performance of each model in the ensemble is plotted as a single line. Performance of the persistence forecast is shown for reference. For the training dataset, the LSTM ensemble consistently outperforms the reference forecast.

dataset, although some ensemble members overestimate low peaks e.g. in June 2011. As expected, RMSE increases with lead time, but precision and recall remain roughly the same (Fig. 6). The curves of precision and recall show some irregularities,

because the total number of events is quite small – only 40 days in a 7-year dataset. This model is able to create a 7-day forecast in 13 seconds on an Intel 5-core processor. The 15-member ensemble shown here takes 3 minutes to run on the same computer. For comparison, the 1D hydraulic model set up for this area, SOBEK 3 (Deltares, 2019) takes 8 minutes to make a 7-day forecast of the Rhine-Meuse estuary.

### 3.2 Model performance on test dataset

Figures 7(a,c,e) show forecasts made for the year 2018, the first year of the test dataset (2018–2020). Forecast values resemble observed concentrations closely for background concentrations (<150 mg l$^{-1}$), with RMSE below 20 mg l$^{-1}$. However, the highest peaks ([Cl] >1000 mg l$^{-1}$) are often underestimated and the lower peaks are often overestimated, which accounts for the higher RMSE for the whole dataset (Fig. 8). This is confirmed by Fig. 7(b,d,f). RMSE increases with lead time, while precision and recall decrease (Fig. 8). Forecast quality decreases fast as lead time increases from 1 to 3 days, but decreases

more slowly after that (compare Figs. 7 and 8). The general tendency of the LSTM models to underestimate peaks leads to higher precision, but sometimes lower recall than the persistence forecast. In terms of RMSE, the LSTM outperforms the persistence forecast from $t_{+2}$ onwards. The RMSE is a factor 4–6 higher for the test dataset than for the training dataset; precision is lower from $t_{+3}$ onwards and recall is significantly lower.

### 3.3 Sensitivity analysis

The sensitivity analysis yields a positive correlation between past and predicted chloride concentrations (Fig. 9), indicating a certain persistence of the pre-existing situation. Furthermore, chloride concentration has a strong negative correlation with

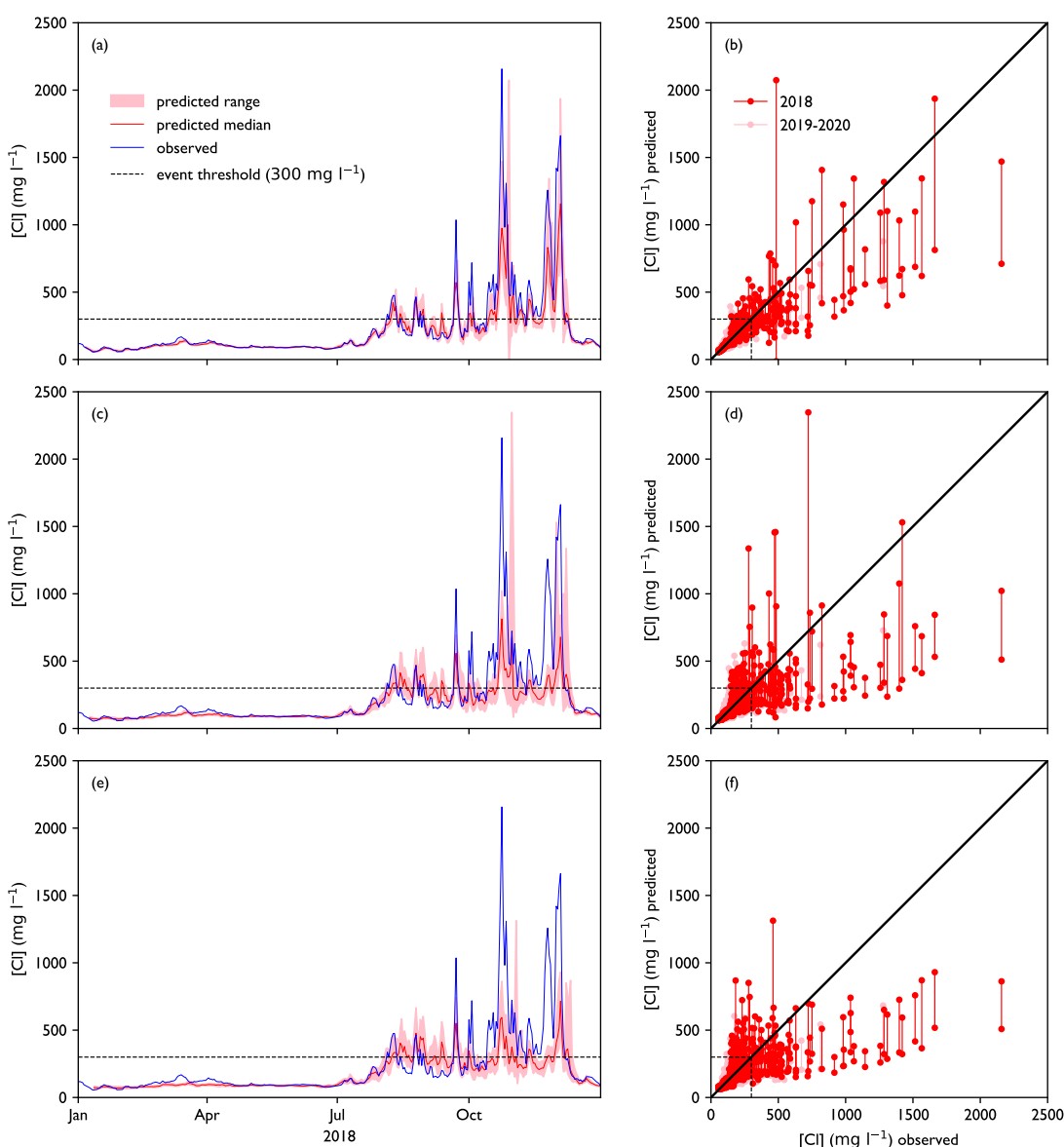

**Figure 7.** Model performance for the test period (2018–2020). Panels (a), (c) and (e) show a collection of forecasts of ClKr400Mean made with lead times of (a) 1 day, (c) 4 days and (e) 7 days for the year 2018, along with observed values. The predicted value is given as an ensemble prediction for each day of the year, created with the lead time mentioned. Median and range of the ensemble prediction are shown. Panels (b), (d) and (f) show predicted vs. observed values for the full test dataset with lead times of 1, 4 and 7 days, respectively. For each day in the training dataset, a vertical line indicates the range of predicted values by the ensemble members. A figure showing all ensemble members separately can be found in Fig. D2.

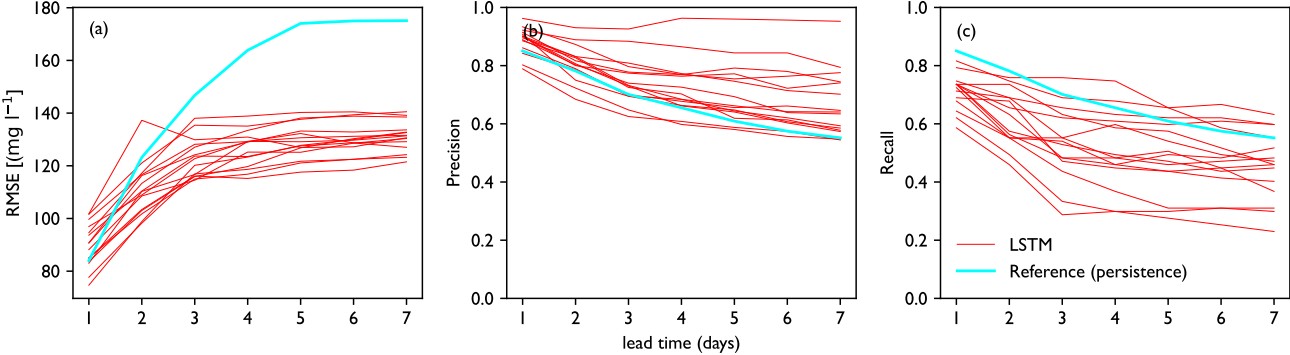

**Figure 8.** Performance metrics for the test period. Panels show (a) RMSE, (b) precision and (c) recall. Performance of each model in the ensemble is plotted as a single line. Performance of the persistence forecast is shown for reference. Compared to the reference, the model is on average closer to the observed concentration (a) and when events are predicted, these usually occur in practice (b). However, many of the LSTM models miss many events compared to the reference (c).

discharge at Lobith and Tiel and a strong positive correlation with water level at Hoek van Holland. This confirms the general understanding that the position of a salt wedge is determined by tidal motion and river discharge. There is also a small negative correlation with water level at Dordrecht and a small positive correlation with discharge at Hagestein. Only a slight positive
correlation is found with southerly and westerly wind speed, even though the results of the Boruta analysis (Sect. 2.3.2) show a less good fit is achieved if it were left out.

## 4 Discussion

### 4.1 Interpretation of results

The results in Sect. 3.1 indicate that it is possible to create an LSTM model to predict chloride concentrations at Krimpen
aan den IJssel. With the current set of variables, we were able to come close to an optimal set of hyperparameters, as can be seen from the good fit of the predictions to the training dataset. However, performance on the test dataset is less good. As can be seen from the results in Sect. 3.2, (Fig. 7), most LSTM models tend to underestimate especially the largest peaks in mean daily chloride concentration. Adjustment of the hyperparameters did not enable us to capture these peaks better. In addition, peaks of intermediate height are frequently overestimated, although the error in that case is smaller than for the very high
peaks. For operational water management, the error in the intermediate peaks is likely to have more consequences than the error in the largest peaks, since these intermediate chloride concentrations make up the transition from a normal situation to a situation where water managers might need to intervene on a larger scale than just closing an inlet for a brief period of time. The threshold value of 300 mg l$^{-1}$ has been chosen to reflect such situations. We see indeed that values of precision and recall are both affected by the errors in peak prediction (Fig. 8). Recall is affected more than precision, which is in line with the

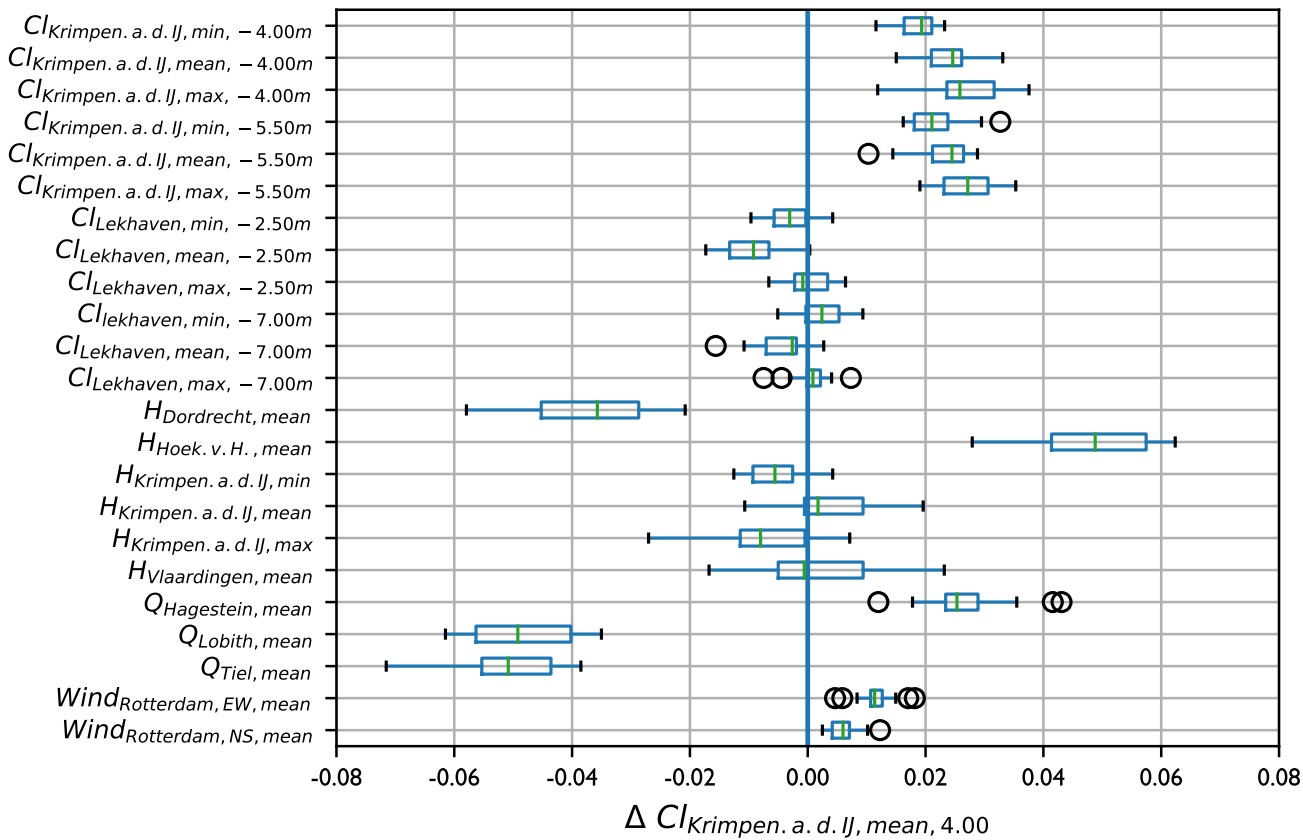

**Figure 9.** Results of the sensitivity analysis. Each row indicates the yearly average increase in daily mean chloride concentration at Krimpen aan den IJssel -4.00 m a.m.s.l. when the input parameter indicated in the graph is increased by 0.2 normalized units. Boxes and whiskers show the ensemble spread and green bars show the ensemble median. Input variables are chloride concentration (Cl), water level (H), discharge (Q) and wind speed (Wind). Subscripts indicate the location (with Krimpen.a.d.IJ = Krimpen aan den IJssel and Hoek.v.H. = Hoek van Holland), daily statistic, depth below mean sea level (chloride only) and direction (wind only, EW = east-west, NS = north-south). Variables are explained in Table 1. All values are expressed in normalized units.

general tendency for underestimation. This means that by relying completely on this model, water managers would be more likely to miss a problematic situation than to take unnecessary action. This is probably not desirable, as the consequences of a missed event are typically more problematic than the consequences of a false alarm (Warmink et al., 2017).

Most results of the sensitivity analysis (Sect. 3.3, Fig. 9) are in line with general expectations of this river system. Higher river discharge dilutes the salt water present at Krimpen aan den IJssel and pushes the salt wedge back towards the sea, whereas
higher sea levels increase the potential for salt water to intrude landward (Savenije, 2012; Sun et al., 2020). This is in line with the findings of Cai et al. (2015, e.g.) and Liu et al. (2017), who find that the salt intrusion length in the Yangtze river is proportional to tidal amplitude at the seaward boundary and inversely proportional to river discharge. Shaha et al. (2013) find a similar dominance of tidal range and discharge for the Sumjin River estuary. The negative correlation between chloride concentration at Krimpen aan den IJssel and water level at Dordrecht seems to indicate that high water levels at Dordrecht
are associated with increasing flow from the Beneden-Merwede through the Noord towards the Nieuwe Maas (Fig. 1). The positive correlation of chloride concentration and discharge at Hagestein is somewhat surprising, as it is mostly a component of discharge at Lobith. Indeed, most chloride peaks coincide with periods of low discharge at Hagestein. However, Hagestein is a managed location with a weir that plays a role in dividing discharge over the Rhine branches. In periods of drought, when chloride concentrations have already started rising, water is sometimes diverted through the Nederrijn/Lek branch, causing
discharge at Hagestein to be relatively high with respect to discharge at Lobith (Hydrologic et al., 2015). We therefore suggest that the models have captured a positive correlation between chloride concentrations and the fraction of Rhine water that flows through the Nederrijn/Lek branch. The very small positive correlation between chloride and southerly and westerly wind speed confirms the observations in Sect. 2.2.2 and Fig. 2, where we also found no consistent relation between wind speed and chloride concentration. This shows that wind speed on its own does not make a difference, but may still influence chloride
concentrations through its interactions with other variables. When we compare this to the work of Xue et al. (2009), who found that wind direction plays a significant role in the salinity distribution of the Changjiang estuary, we find that the specific shape of a delta can influence the importance of various variables, a phenomenon that is also suggested by the idealized study of Jongbloed et al. (2022).

## 4.2 Limitations and outlook

The shorter runtime of the machine learning model allows users to run simulations as an ensemble well ahead of time. The current version of the model only works for one location, but if we can capture multiple locations in the delta while keeping runtime in the same order of magnitude, we can do many more predictions, including simulations of extreme scenarios. The larger number of scenarios that can be investigated gives water managers more opportunities to take mitigating measures and ensure freshwater availability. It is difficult to simulate the effect of management decisions, because they are not included
explicitly in the training dataset. However, if the effect of a management decision on water level and discharge is known, the simulated water level and discharge could be used as inputs for the model. We still have to implement the model in a forecasting system to assess how well this would work.

Machine learning models are known to have their limits when it comes to forecasting extreme events: since these events are rare by nature, a model that is trained on a long timeseries will have far more examples of regular than of extreme conditions (Carbajal and Bellos, 2018). The model is therefore likely more skillful in forecasting baselines than in forecasting (extreme) peaks. We observe this phenomenon in our results for the test dataset (Sect. 3.2). As 2018 was a very dry year (Buitink et al., 2020), the chloride concentrations reached levels that had not been observed in our training dataset. Our model was therefore less skillful than desired at predicting especially the highest peaks (>1000 mg l$^{-1}$). This is a problematic situation, since climate change is expected to increase sea level and decrease river discharge in spring, summer and fall for our study area, which makes the occurrence of such peaks more likely (Lenderink and Beersma, 2015; Van den Brink et al., 2019). If used for operational forecasting, this model is therefore likely to deal with unprecedented situations more frequently in the future. To tackle this issue, we would propose to update the training dataset and retrain the model yearly, adding new, possibly more extreme, observations to the record and make the model better suited to forecast extreme situations in the future.

Another application of machine learning is to use a machine learning model as a model emulator (Carbajal and Bellos, 2018). In this application, another model, often a physical model, is run with a large range of conditions, and its results are then supplied to a machine learning algorithm. The algorithm learns the original model's behaviour in terms of inputs and outputs, without considering its internal mechanics. This yields a simplified model, which may not be as accurate as the original, especially in unprecedented situations, but which tends to be much faster (Silva et al., 2021; Gettelman et al., 2021). Having identified the input and output variables needed to set up a salt intrusion model, we could train a similar model with the input and output variables of a three-dimensional model of the Rhine-Meuse delta, which is currently under development. This would allow us to mimic a wide range of possible conditions with the machine learning model, without having to resort to extrapolation. If successful, the result would be an approximation of a physical model that can run fast enough for operational use, making it suitable for interactive simulations in, for instance, a serious game or digital twin. In a serious game, a study area is represented in a board or computer game environment, which allows stakeholders to try out different strategies in a safe environment, learning about each other's interests. A digital twin has some of these characteristics as well, but aims to be more realistic and usable in real time, which makes it more suitable as a simulation and management tool. A model emulator could also be useful to simulate extreme situations that are underrepresented in the existing record, as discussed in the previous paragraph.

Our results show that a reasonable prediction of chloride concentrations up to seven days ahead can be achieved at one location using this model: although the error in peak height is quite large, timing and occurrence of peaks are well-captured. We therefore expect that a similar model setup can be successful for other locations in the delta for which salt intrusion is a similar threat to freshwater availabiltiy, such as the junction of the Oude Maas and Spui and the confluence of the Noord and Lek (Fig. 1) (Van den Brink et al., 2019). Extending the analysis is likely to teach us more about the dependencies in this system, which can in turn help to improve the existing model.

To improve the model, we are also considering adding some physical constraints, such as mass balances in a control section. Bertels and Willems (2023) applied such an approach to the Scheldt estuary and achieved notably better results than with a

purely data-driven model. However, the large number of branches in the Rhine-Meuse delta may make application of a mass balance approach quite challenging.

In Sect. 3.1 we show that this model is much faster than the 1D physical model SOBEK. On the other hand, SOBEK is
365 run for the entire delta area, whereas the machine learning model focusses on a single output location. When the model is extended to comprise multiple locations within the study area, runtime will increase. However, if we focus on a limited number of stations, we still expect the machine learning model to be significantly faster than the physical model, since the machine learning model performs forward calculations rather than solving differential equations.

In the current model setup, water levels, discharges and wind speeds at $t_{+1}$ are used to forecast chloride concentrations at
370 $t_{+1}$. In an actual operational setting, these values would be retrieved from other models, with their own uncertainties. These uncertainties then propagate to the chloride concentration forecast. In our analysis, we used a historical dataset to fit and test the model, using the actual observations at $t_{+1}$. This way we have uniform data of a constant uncertainty with which we can derive and evaluate a model. However, this also means that the model's performance as described in Section 3 is higher than it would be in an actual operational context. Setting the model up to function in a forecasting system, using the outputs of other
models as inputs, is a follow-up step in our research.

In this study, we have made forecasts of daily mean, minimum and maximum chloride concentrations. We have chosen daily values to limit error accumulation when creating a 7-day forecast. However, there are many regions where operational water managers need predictions with a higher temporal resolution, e.g. to determine at what time of day certain inlets should be opened or closed (Pezij et al., 2019; Tian, 2015). We will therefore attempt to train the model to make predictions on shorter
timescales, for which other variables might be needed than the ones we used in this analysis.

This model was developed for a delta with a complex geography. Nevertheless, we could develop a data-driven model with a total of 12 input variables (counting the minimum, mean and maximum as features of a single variable, and doing the same for the east-west and north-south component of the wind speed). We could have added more variables, but timeseries exploration and Boruta analysis showed that these would be redundant. We can therefore conclude that the number of measurement stations
needed to train a model like this is not very high. A sufficiently long record with few gaps remains needed, however. With a training period of seven years, satisfactory results can be achieved. We therefore suggest that this approach can be extended to other deltas where an adequate measurement setup exists or where it is being developed. Especially in deltas with a single branch, a smaller number of stations would probably suffice, although it is important that the location of the stations does not change. Since a machine learning approach does not require a full understanding of the system's internal mechanics but relies
on patterns in the data, it should not be a problem if the system functions somewhat differently than the one we studied. For example, a study of the Merrimack river revealed a similar dependency on wind, discharge and seawater level, although the relative contributions of these factors were different (Ralston et al., 2010). Therefore, we expect that most of the features used for this model can be applied in other study areas. However, it is possible that other quantities, such as precipitation sums or offshore water levels, need to be added to obtain a satisfactory solution. The timescale considered may also play a role for
the variables that are needed. For example, a study of the La Comté river showed that a 3-hour prediction of salinity can be

made with just seawater level and river discharge (Rohmer and Brisset, 2017). On the other hand, Lu et al. (2021) found a dependency on the number of sunspots when analyzing salt intrusion in the Pearl River delta on a monthly timescale.

## 5   Conclusions

We used a machine learning approach with a Long Short-Term Memory network to set up a data-driven model for forecasting chloride concentrations at Krimpen aan den IJssel, located in the Rhine-Meuse delta. Using observations of chloride concentration, water level, discharge and wind speed at a total of 9 locations, we were able to forecast daily minimum, mean and maximum chloride concentrations up to 7 days ahead. The baseline concentrations (<150 mg l$^{-1}$) are predicted well by this model (RMSE < 20 mg l$^{-1}$). Timing of chloride peaks is also predicted well, but their magnitude is underestimated. This deviation increases quite fast between lead times of 1 and 4 days, and more slowly at even longer lead times. A sensitivity analysis shows a positive correlation with antecedent chloride concentrations and seawater level and a negative correlation with discharge through the main river branches. We expect that the quality of this model can be improved with lessons learned at other locations, which will allow us to construct a more comprehensive forecasting tool for the Rhine-Meuse delta. A similar approach is likely to be successful for other deltas, especially those that have a comparable or simpler geography than our study area.

*Code and data availability.* Data and software are available on the 4TU repository, using https://doi.org/10.4121/21944249 for the data and https://doi.org/10.4121/21946724 for the software. Raw data can be found on https://www.knmi.nl/nederland-nu/ klimatologie/daggegevens and https://waterinfo.rws.nl/#!/kaart/Waterbeheer/. The software will undergo further development in future. The most recent version can be found on https://github.com/BasWullems/salt_intrusion_lstm.

## Appendix A:   Correlation of water levels

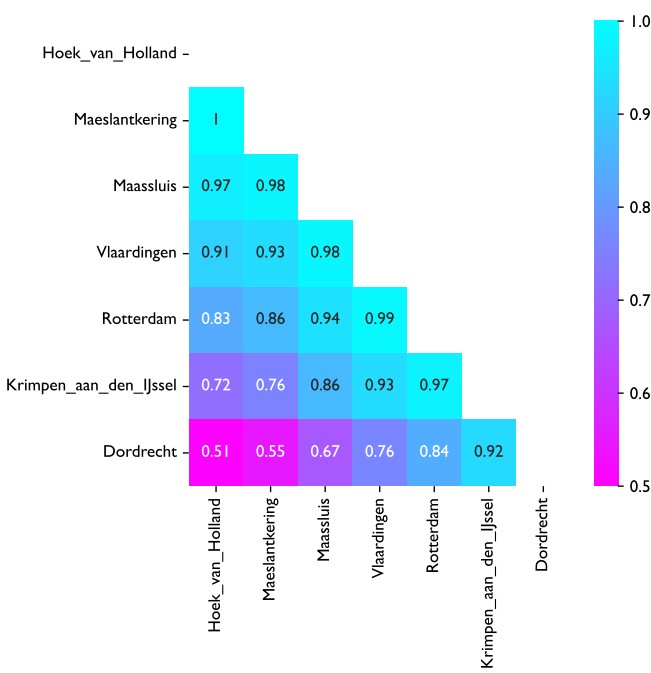

**Figure A1.** Pearson correlation coefficient for water levels in the study area.

Fig. A1 shows the Pearson correlation values between water levels at several locations in the study area. It is visible that water levels at Dordrecht deviate from the other locations. Water levels at Hoek van Holland, Measlantkering, Maassluis, Vlaardingen and Rotterdam are very similar, which is why we only retain Hoek van Holland and Vlaardingen.

## Appendix B: Boruta analysis

As mentioned in sect. 2.3.2, we ran the Boruta feature selection algorithm (Homola et al., 2022) on the daily minimum, mean

and maximum chloride concentration at Krimpen aan den IJssel for $t_{+1}$. We started by creating a random forest model to predict the output feature. Then, in each iteration, one of the indicated features (e.g. daily maximum water level at Hoek van Holland at $t_{-6}$) was randomly shuffled, and a new model was derived. Performance of this new model (mean squared error) was then compared to that of the original model. If the new model performed worse than the original model, shuffling the input feature created a significant amount of noise, indicating this feature is important for prediction of the output feature. Making

this comparison for every input feature, the algorithm then ranked the features as 1 (important), 2 (moderately important or inconclusive), or anything higher (unimportant). We show this in Tables B1, B2 & B3 as + (important) o (moderately important or inconclusive) and -(unimportant). The algorithm was run with a significance level of 0.05 and a target to retain 70% of features.

    Comparing Tables B1, B2 & B3, no single cutoff point between important and unimportant features emerges. However,

based on the number of important timesteps for each variable, we draw the following conclusions:

- No more than 5 timesteps of any variable are needed to make a prediction for $t_{+1}$.

- $t_{-6}$ is often found to be important, while $t_{-5}$ and $t_{-4}$ is often found to be inconclusive. We expect the number of timesteps used to determine a trend to be more important than the actual timestep and therefore choose a continuous series of measurements from $t_{-4}$ to $t_0$.

- Minimum, mean and maximum chloride concentration at Krimpen aan den IJssel and Lekhaven are all important to predict any of the output features.

- Minimum, mean and maximum water levels at Krimpen aan den IJssel are important to predict chloride concentrations. For the other stations, one daily statistic of water level appears to suffice, so we only keep the daily mean.

- All discharge stations are important to predict chloride concentrations.

- Daily mean wind speed in both directions is important to predict chloride concentrations.

    Based on these conclusions, we selected the features in Table 1.

## Appendix C: Hyperparameter tuning overview

**Table B1.** Results of the Boruta feature selection process for daily minimum chloride concentration at Krimpen aan den IJssel at $t_{+1}$. A + indicates that the indicated feature is important to predict the the output feature, while a - indicates that it is not important. An o indicates that the importance of the feature is moderate or uncertain. An n indicates that the feature is not applicable as an input feature.

| Variable | Timestep | | | | | | | | No. relevant feautures |
| --- | --- | --- | --- | --- | --- | --- | --- | --- | --- |
| | -6 | -5 | -4 | -3 | -2 | -1 | 0 | +1 | |
| ClKr400Min | + | - | 0 | + | + | + | + | n | 5 |
| ClKr400Mean | + | - | + | - | 0 | - | + | n | 3 |
| ClKr400Max | + | 0 | - | + | + | + | + | n | 5 |
| ClKr550Min | - | - | + | 0 | 0 | 0 | + | n | 2 |
| ClKr550Mean | - | - | - | 0 | 0 | 0 | + | n | 1 |
| ClKr550Max | 0 | 0 | - | - | + | 0 | + | n | 2 |
| ClLkh250Min | + | - | + | + | + | 0 | 0 | n | 4 |
| ClLkh250Mean | - | - | - | - | - | + | - | n | 1 |
| ClLkh250Max | + | - | - | 0 | - | - | + | n | 2 |
| ClLkh500Min | + | - | + | + | 0 | 0 | 0 | n | 3 |
| ClLkh500Mean | + | - | - | - | - | - | + | n | 2 |
| ClLkh500Max | + | - | 0 | + | 0 | - | + | n | 3 |
| ClLkh700Min | + | - | + | + | - | + | + | n | 5 |
| ClLkh700Mean | 0 | - | - | - | - | - | + | n | 1 |
| ClLkh700Max | - | + | 0 | - | 0 | - | + | n | 2 |
| HDrdMin | - | - | - | 0 | - | + | + | + | 3 |
| HDrdMean | + | - | + | 0 | - | + | + | + | 5 |
| HDrdMax | - | - | 0 | - | - | 0 | + | + | 2 |
| HHvhMin | - | - | - | - | - | + | + | 0 | 2 |
| HHvhMean | - | - | - | + | + | + | + | + | 5 |
| HHvhMax | - | - | - | - | + | - | + | + | 3 |
| HKrMin | - | - | - | 0 | - | - | - | + | 1 |
| HKrMean | - | - | 0 | 0 | - | + | - | + | 2 |
| HKrMax | - | 0 | - | - | + | 0 | 0 | 0 | 1 |
| HVlaMin | - | - | - | - | - | - | + | - | 1 |
| HVlaMean | - | - | 0 | + | - | + | - | 0 | 2 |
| HVlaMax | - | - | - | - | - | - | - | 0 | 0 |
| QHagMean | + | 0 | - | 0 | + | + | + | + | 5 |
| QLobMean | 0 | - | + | + | + | + | + | + | 6 |
| QTielMean | + | - | - | + | + | + | + | + | 6 |
| WindEW | - | 0 | - | - | - | - | + | - | 1 |
| WindNS | - | - | - | - | + | + | - | - | 2 |

**Table B2.** Results of the Boruta feature selection process for daily mean chloride concentration at Krimpen aan den IJssel at $t_{+1}$. A + indicates that the indicated feature is important to predict the the output feature, while a - indicates that it is not important. An o indicates that the importance of the feature is moderate or uncertain. An n indicates that the feature is not applicable as an input feature.

| Variable | Timestep | | | | | | | | No. relevant feautures |
|---|---|---|---|---|---|---|---|---|---|
| | -6 | -5 | -4 | -3 | -2 | -1 | 0 | +1 | |
| ClKr400Min | + | - | + | - | - | 0 | + | n | 3 |
| ClKr400Mean | + | + | + | - | 0 | + | + | n | 5 |
| ClKr400Max | + | - | - | - | 0 | + | + | n | 3 |
| ClKr550Min | - | 0 | - | - | - | - | + | n | 1 |
| ClKr550Mean | 0 | - | 0 | 0 | - | + | + | n | 2 |
| ClKr550Max | - | - | - | 0 | 0 | + | + | n | 2 |
| ClLkh250Min | 0 | + | - | + | + | + | + | n | 5 |
| ClLkh250Mean | - | - | - | - | - | + | 0 | n | 1 |
| ClLkh250Max | - | - | - | - | - | + | + | n | 2 |
| ClLkh500Min | - | - | - | - | 0 | 0 | + | n | 1 |
| ClLkh500Mean | - | - | - | - | - | - | + | n | 1 |
| ClLkh500Max | - | - | - | - | + | + | + | n | 3 |
| ClLkh700Min | - | - | + | - | + | + | + | n | 4 |
| ClLkh700Mean | - | - | + | + | - | + | + | n | 4 |
| ClLkh700Max | - | + | - | - | 0 | + | + | n | 3 |
| HDrdMin | - | - | 0 | - | + | + | + | + | 4 |
| HDrdMean | - | 0 | - | + | - | + | + | + | 4 |
| HDrdMax | - | - | 0 | 0 | - | + | + | + | 3 |
| HHvhMin | - | - | - | - | - | + | + | + | 3 |
| HHvhMean | - | - | - | - | - | + | + | + | 3 |
| HHvhMax | - | - | - | - | - | 0 | + | + | 2 |
| HKrMin | - | - | - | - | - | + | - | + | 2 |
| HKrMean | - | + | - | + | + | + | - | + | 5 |
| HKrMax | - | - | - | - | 0 | 0 | - | + | 1 |
| HVlaMin | - | - | - | - | - | - | + | + | 2 |
| HVlaMean | - | - | - | - | - | + | + | + | 3 |
| HVlaMax | - | - | - | - | - | + | 0 | + | 2 |
| QHagMean | - | + | + | - | - | + | 0 | 0 | 3 |
| QLobMean | - | - | - | 0 | + | + | + | + | 4 |
| QTielMean | + | 0 | + | + | + | + | + | + | 7 |
| WindEW | + | + | - | - | - | - | + | + | 4 |
| WindNS | - | - | - | - | - | 0 | + | + | 2 |

**Table B3.** Results of the Boruta feature selection process for daily maximum chloride concentration at Krimpen aan den IJssel at $t_{+1}$. A + indicates that the indicated feature is important to predict the the output feature, while a - indicates that it is not important. An o indicates that the importance of the feature is moderate or uncertain. An n indicates that the feature is not applicable as an input feature.

| Variable | Timestep | | | | | | | | No. relevant feautures |
|---|---|---|---|---|---|---|---|---|---|
| | -6 | -5 | -4 | -3 | -2 | -1 | 0 | +1 | |
| ClKr400Min | + | 0 | - | - | + | - | 0 | n | 2 |
| ClKr400Mean | + | - | - | 0 | 0 | + | + | n | 3 |
| ClKr400Max | + | + | - | - | 0 | + | + | n | 4 |
| ClKr550Min | 0 | - | - | - | - | - | - | n | 0 |
| ClKr550Mean | 0 | - | - | 0 | - | - | + | n | 1 |
| ClKr550Max | - | - | - | - | - | 0 | + | n | 1 |
| ClLkh250Min | + | + | 0 | 0 | 0 | + | + | n | 4 |
| ClLkh250Mean | 0 | - | - | - | 0 | + | + | n | 2 |
| ClLkh250Max | - | - | - | - | 0 | + | + | n | 2 |
| ClLkh500Min | - | - | - | - | + | 0 | + | n | 2 |
| ClLkh500Mean | 0 | - | - | - | - | 0 | + | n | 1 |
| ClLkh500Max | - | - | - | - | + | + | + | n | 3 |
| ClLkh700Min | + | + | + | + | + | + | + | n | 7 |
| ClLkh700Mean | - | 0 | - | + | - | + | + | n | 3 |
| ClLkh700Max | - | + | + | + | - | + | + | n | 5 |
| HDrdMin | - | - | + | - | - | 0 | 0 | + | 2 |
| HDrdMean | - | - | - | + | + | 0 | + | + | 4 |
| HDrdMax | - | - | - | - | 0 | + | 0 | + | 2 |
| HHvhMin | - | + | 0 | - | - | - | + | + | 3 |
| HHvhMean | + | - | - | - | - | + | + | + | 4 |
| HHvhMax | + | - | - | 0 | - | + | - | + | 3 |
| HKrMin | - | - | - | - | 0 | - | - | - | 0 |
| HKrMean | - | - | - | - | 0 | - | - | + | 1 |
| HKrMax | - | - | - | - | 0 | + | - | + | 2 |
| HVlaMin | - | - | - | - | 0 | - | + | + | 2 |
| HVlaMean | - | - | - | - | - | - | - | + | 1 |
| HVlaMax | - | - | - | - | - | + | - | + | 2 |
| QHagMean | - | + | + | 0 | 0 | + | + | + | 5 |
| QLobMean | - | - | - | 0 | + | + | + | + | 4 |
| QTielMean | + | + | - | + | + | + | + | + | 7 |
| WindEW | + | - | - | - | - | - | + | + | 3 |
| WindNS | 0 | - | - | - | - | + | 0 | + | 2 |

**Table C1.** Hyperparameters that were tried in the model tuning process described in Sect. 2.3.5. Each model number corresponds to a set of hyperparameters with which three randomly initialized model were trained. The basic structure of the model is given in Table 4. The column 'output weighted' indicates the weight given to the following output parameters: ClKr400Min, ClKr400Mean, ClKr400Max, ClKr550Min, ClKr550Mean, ClKr550Max, ClLkh250Min, ClLkh250Mean, ClLkh250Max, ClLkh700Min, ClLkh700Mean, ClLkh700Max. Definitions of these variables are found in Table 1. Results of the training run are in Tables C2, C3, C4 & C5. Model 21 was selected as the best performing model (Table 2).

| Model no. | Size LSTM 1 | Size LSTM 2 | Batch size | Extra dense layer size | Dropout (after LSTM) | Dropout (after dense) | Output weighted |
|---|---|---|---|---|---|---|---|
| 1 | 32 | 32 | 16 | 16 | 0.2 | 0.2 | NO |
| 2 | 32 | 32 | 32 | 16 | 0.2 | 0.2 | NO |
| 3 | 32 | 32 | 64 | 16 | 0.2 | 0.2 | NO |
| 4 | 32 | 32 | 64 | 0 | 0.2 | 0 | NO |
| 5 | 32 | 32 | 64 | 32 | 0.2 | 0.2 | NO |
| 6 | 16 | 16 | 64 | 16 | 0.2 | 0.2 | NO |
| 7 | 16 | 16 | 64 | 32 | 0.2 | 0.2 | NO |
| 8 | 16 | 16 | 64 | 32 | 0.2 | 0.2 | 2,3,2,1,1,1,1,1,1,1,1 |
| 9 | 16 | 16 | 64 | 32 | 0.2 | 0.2 | 1,2,2,1,1,1,1,1,1,1,1 |
| 10 | 16 | 16 | 32 | 32 | 0.2 | 0.2 | 1,2,2,1,1,1,1,1,1,1,1 |
| 11 | 32 | 32 | 32 | 32 | 0.2 | 0.2 | 1,2,2,1,1,1,1,1,1,1,1 |
| 12 | 32 | 32 | 64 | 32 | 0.2 | 0.2 | 1,2,2,1,1,1,1,1,1,1,1 |
| 13 | 16 | 16 | 32 | 32 | 0.2 | 0.2 | 2,2,2,1,1,1,1,1,1,1,1 |
| 14 | 16 | 16 | 32 | 32 | 0.2 | 0.2 | 2,2,3,1,1,1,1,1,1,1,1 |
| 15 | 32 | 32 | 32 | 32 | 0.2 | 0.2 | 2,2,3,1,1,1,1,1,1,1,1 |
| 16 | 32 | 32 | 32 | 32 | 0.2 | 0.2 | 2,3,2,1,1,1,1,1,1,1,1 |
| 17 | 32 | 32 | 32 | 32 | 0.2 | 0.2 | 2,3,2,1,2,1,1,1,1,1,1 |
| 18 | 32 | 32 | 32 | 32 | 0.2 | 0.2 | 2,3,3,1,2,1,1,1,1,1,1 |
| 19 | 32 | 32 | 32 | 0 | 0.2 | 0 | 2,3,3,1,2,1,1,1,1,1,1 |
| 20 | 32 | 32 | 32 | 0 | 0.3 | 0 | 2,3,3,1,2,1,1,1,1,1,1 |
| *21* | *32* | *32* | *64* | *0* | *0.3* | *0* | *2,3,3,1,2,1,1,1,1,1,1* |
| 22 | 32 | 32 | 16 | 0 | 0.3 | 0 | 2,3,3,1,2,1,1,1,1,1,1 |
| 23 | 32 | 32 | 16 | 32 | 0.3 | 0.3 | 2,3,3,1,2,1,1,1,1,1,1 |
| 24 | 32 | 32 | 16 | 32 | 0.3 | 0 | 2,3,3,1,2,1,1,1,1,1,1 |
| 25 | 32 | 32 | 16 | 32 | 0 | 0.3 | 2,3,3,1,2,1,1,1,1,1,1 |
| 26 | 32 | 32 | 16 | 32 | 0.5 | 0.5 | 2,3,3,1,2,1,1,1,1,1,1 |
| 27 | 32 | 32 | 32 | 32 | 0.5 | 0.5 | 2,3,3,1,2,1,1,1,1,1,1 |
| 28 | 32 | 32 | 32 | 32 | 0.5 | 0 | 2,3,3,1,2,1,1,1,1,1,1 |
| 29 | 64 | 64 | 32 | 64 | 0.5 | 0.5 | 2,3,3,1,2,1,1,1,1,1,1 |
| 30 | 64 | 64 | 32 | 0 | 0.5 | 0 | 2,3,3,1,2,1,1,1,1,1,1 |
| 31 | 128 | 128 | 32 | 0 | 0.5 | 0 | 2,3,3,1,2,1,1,1,1,1,1 |
| 32 | 128 | 128 | 32 | 0 | 0.2 | 0 | 2,3,3,1,2,1,1,1,1,1,1 |
| 33 | 32 | 32 | 32 | 0 | 0.5 | 0 | 2,3,3,1,2,1,1,1,1,1,1 |
| 34 | 32 | 32 | 64 | 0 | 0.5 | 0 | 2,3,3,1,2,1,1,1,1,1,1 |
| 35 | 32 | 32 | 16 | 0 | 0.5 | 0 | 2,3,3,1,2,1,1,1,1,1,1 |
| 36 | 32 | 32 | 16 | 0 | 0.2 | 0 | 2,3,3,1,2,1,1,1,1,1,1 |
| 37 | 32 | 32 | 64 | 0 | 0.2 | 0 | 2,3,3,1,2,1,1,1,1,1,1 |

**Table C2.** RMSE for three randomly initialized models per hyperparameter setup. Every run shows the results of one model. Model numbers correspond to the hyperparameter sets in Table C1. RMSE was calculated for three lead times: $t_{+1}$, $t_{+4}$ and $t_{+7}$. RMSE can range from 0 (perfect) to infinity (wrong).

| Model no. | RMSE (mg l$^{-1}$) | | | | | | | | |
|---|---|---|---|---|---|---|---|---|---|
| | Run 1 | | | Run 2 | | | Run 3 | | |
| | $t_{+1}$ | $t_{+4}$ | $t_{+7}$ | $t_{+1}$ | $t_{+4}$ | $t_{+7}$ | $t_{+1}$ | $t_{+4}$ | $t_{+7}$ |
| 1 | 19 | 26 | 27 | 24 | 32 | 34 | 20 | 27 | 28 |
| 2 | 20 | 28 | 29 | 19 | 25 | 28 | 19 | 26 | 28 |
| 3 | 19 | 25 | 26 | 20 | 26 | 28 | 20 | 28 | 31 |
| 4 | 20 | 26 | 28 | 18 | 24 | 27 | 19 | 25 | 28 |
| 5 | 19 | 23 | 25 | 20 | 26 | 29 | 22 | 28 | 30 |
| 6 | 23 | 29 | 32 | 24 | 30 | 36 | 23 | 30 | 33 |
| 7 | 23 | 32 | 35 | 20 | 28 | 31 | 23 | 28 | 30 |
| 8 | 20 | 26 | 29 | 21 | 31 | 33 | 22 | 28 | 31 |
| 9 | 20 | 26 | 28 | 22 | 28 | 31 | 19 | 26 | 28 |
| 10 | 23 | 29 | 30 | 21 | 27 | 29 | 19 | 27 | 29 |
| 11 | 19 | 25 | 26 | 18 | 27 | 29 | 18 | 23 | 25 |
| 12 | 19 | 27 | 28 | 18 | 24 | 27 | 21 | 30 | 32 |
| 13 | 21 | 30 | 32 | 21 | 31 | 32 | 21 | 29 | 30 |
| 14 | 20 | 28 | 30 | 21 | 29 | 31 | 20 | 27 | 30 |
| 15 | 18 | 25 | 27 | 18 | 23 | 25 | 17 | 22 | 24 |
| 16 | 21 | 26 | 28 | 18 | 26 | 28 | 19 | 27 | 30 |
| 17 | 21 | 28 | 29 | 21 | 28 | 29 | 19 | 25 | 27 |
| 18 | 20 | 27 | 28 | 18 | 26 | 28 | 19 | 26 | 27 |
| 19 | 18 | 25 | 27 | 18 | 25 | 27 | 20 | 27 | 29 |
| 20 | 21 | 28 | 31 | 20 | 27 | 28 | 18 | 25 | 26 |
| *21* | *19* | *24* | *26* | *19* | *27* | *28* | *18* | *23* | *24* |
| 22 | 21 | 28 | 30 | 20 | 27 | 29 | 18 | 24 | 25 |
| 23 | 20 | 27 | 29 | 19 | 26 | 28 | 21 | 28 | 29 |
| 24 | 18 | 25 | 27 | 17 | 24 | 26 | 19 | 24 | 26 |
| 25 | 19 | 27 | 29 | 21 | 30 | 32 | 17 | 23 | 25 |
| 26 | 25 | 32 | 35 | 27 | 36 | 37 | 24 | 32 | 33 |
| 27 | 22 | 30 | 31 | 22 | 30 | 31 | 22 | 28 | 29 |
| 28 | 17 | 25 | 27 | 20 | 29 | 31 | 21 | 28 | 29 |
| 29 | 18 | 26 | 27 | 20 | 26 | 28 | 20 | 26 | 29 |
| 30 | 19 | 25 | 27 | 20 | 25 | 26 | 18 | 24 | 25 |
| 31 | 18 | 22 | 24 | 17 | 24 | 25 | 18 | 23 | 25 |
| 32 | 19 | 25 | 26 | 19 | 25 | 26 | 18 | 22 | 23 |
| 33 | 21 | 30 | 31 | 21 | 29 | 30 | 20 | 27 | 28 |
| 34 | 22 | 28 | 30 | 22 | 30 | 32 | 20 | 28 | 29 |
| 35 | 20 | 30 | 31 | 20 | 26 | 28 | 19 | 26 | 28 |
| 36 | 19 | 24 | 26 | 21 | 25 | 26 | 18 | 23 | 25 |
| 37 | 19 | 27 | 29 | 20 | 26 | 28 | 20 | 28 | 29 |

**Table C3.** Precision for three randomly initialized models per hyperparameter setup. Every run shows the results of one model. Model numbers correspond to the hyperparameter sets in Table C1. Precision was calculated for three lead times: $t_{+1}$, $t_{+4}$ and $t_{+7}$. Precision can range from 0 (wrong) to 1 (perfect).

| Model no. | Precision | | | | | | | | |
| --- | --- | --- | --- | --- | --- | --- | --- | --- | --- |
| | Run 1 | | | Run 2 | | | Run 3 | | |
| | $t_{+1}$ | $t_{+4}$ | $t_{+7}$ | $t_{+1}$ | $t_{+4}$ | $t_{+7}$ | $t_{+1}$ | $t_{+4}$ | $t_{+7}$ |
| 1 | 0.83 | 0.90 | 0.88 | 0.91 | 0.97 | 0.97 | 0.82 | 0.82 | 0.82 |
| 2 | 0.87 | 0.81 | 0.85 | 0.83 | 0.83 | 0.83 | 0.84 | 0.91 | 0.91 |
| 3 | 0.88 | 0.81 | 0.80 | 0.93 | 0.97 | 0.94 | 0.86 | 0.88 | 0.89 |
| 4 | 0.86 | 0.83 | 0.76 | 0.91 | 0.94 | 0.89 | 0.97 | 0.87 | 0.89 |
| 5 | 0.89 | 0.86 | 0.81 | 0.81 | 0.78 | 0.72 | 0.94 | 0.93 | 0.93 |
| 6 | 0.97 | 0.85 | 0.75 | 0.87 | 0.72 | 0.60 | 0.86 | 0.86 | 0.82 |
| 7 | 0.93 | 0.97 | 0.96 | 0.94 | 0.91 | 0.91 | 0.83 | 0.86 | 0.83 |
| 8 | 0.85 | 0.79 | 0.78 | 0.94 | 0.97 | 0.94 | 0.94 | 0.77 | 0.77 |
| 9 | 0.86 | 0.89 | 0.79 | 0.91 | 0.89 | 0.73 | 0.91 | 0.88 | 0.89 |
| 10 | 0.76 | 0.73 | 0.70 | 0.89 | 0.91 | 0.88 | 0.82 | 0.87 | 0.81 |
| 11 | 0.89 | 0.87 | 0.80 | 0.85 | 0.89 | 0.86 | 0.92 | 0.83 | 0.81 |
| 12 | 0.83 | 0.83 | 0.85 | 0.89 | 0.89 | 0.82 | 0.91 | 0.91 | 0.91 |
| 13 | 0.97 | 1.00 | 1.00 | 0.94 | 0.97 | 0.96 | 0.86 | 0.89 | 0.85 |
| 14 | 0.85 | 0.85 | 0.78 | 0.93 | 0.97 | 1.00 | 0.81 | 0.78 | 0.78 |
| 15 | 0.85 | 0.92 | 0.92 | 0.84 | 0.84 | 0.79 | 0.89 | 0.80 | 0.73 |
| 16 | 0.88 | 0.81 | 0.83 | 0.86 | 0.89 | 0.91 | 0.86 | 0.83 | 0.81 |
| 17 | 0.82 | 0.78 | 0.82 | 0.94 | 0.91 | 0.94 | 0.89 | 0.86 | 0.84 |
| 18 | 0.85 | 0.89 | 0.91 | 0.80 | 0.87 | 0.77 | 0.89 | 0.85 | 0.89 |
| 19 | 0.94 | 0.97 | 0.94 | 0.91 | 0.92 | 0.94 | 0.87 | 0.87 | 0.87 |
| 20 | 0.94 | 0.80 | 0.69 | 0.86 | 0.91 | 0.89 | 0.94 | 0.91 | 0.89 |
| *21* | *0.91* | *0.89* | *0.91* | *0.87* | *0.82* | *0.80* | *0.92* | *0.92* | *0.85* |
| 22 | 0.91 | 0.91 | 0.91 | 0.92 | 0.89 | 0.89 | 0.87 | 0.83 | 0.79 |
| 23 | 0.97 | 0.83 | 0.80 | 0.80 | 0.83 | 0.80 | 0.97 | 0.97 | 0.91 |
| 24 | 0.84 | 0.89 | 0.87 | 0.94 | 0.97 | 0.94 | 0.88 | 0.86 | 0.86 |
| 25 | 0.88 | 0.90 | 0.87 | 0.86 | 0.89 | 0.82 | 0.91 | 0.87 | 0.87 |
| 26 | 0.83 | 0.71 | 0.62 | 0.94 | 0.97 | 0.94 | 1.00 | 0.93 | 0.93 |
| 27 | 0.94 | 0.94 | 0.91 | 0.91 | 0.91 | 0.86 | 0.88 | 0.89 | 0.86 |
| 28 | 0.87 | 0.89 | 0.85 | 0.86 | 0.92 | 0.82 | 0.92 | 0.92 | 0.91 |
| 29 | 0.89 | 0.95 | 0.90 | 0.90 | 0.83 | 0.83 | 0.87 | 0.80 | 0.79 |
| 30 | 0.83 | 0.86 | 0.84 | 0.85 | 0.85 | 0.85 | 0.89 | 0.91 | 0.91 |
| 31 | 0.97 | 0.89 | 0.88 | 0.94 | 0.94 | 0.91 | 0.91 | 0.97 | 0.91 |
| 32 | 0.86 | 0.86 | 0.75 | 0.92 | 0.86 | 0.83 | 0.97 | 1.00 | 1.00 |
| 33 | 0.97 | 0.96 | 0.96 | 0.97 | 0.97 | 0.97 | 0.86 | 0.88 | 0.83 |
| 34 | 0.86 | 0.86 | 0.86 | 0.86 | 0.86 | 0.83 | 0.91 | 0.91 | 0.88 |
| 35 | 0.87 | 0.87 | 0.86 | 0.82 | 0.82 | 0.84 | 0.89 | 0.92 | 0.87 |
| 36 | 0.91 | 0.97 | 0.97 | 0.89 | 0.83 | 0.83 | 0.89 | 0.92 | 0.92 |
| 37 | 0.91 | 0.89 | 0.87 | 0.94 | 0.89 | 0.74 | 0.85 | 0.89 | 0.89 |

**Table C4.** Recall for three randomly initialized models per hyperparameter setup. Every run shows the results of one model. Model numbers correspond to the hyperparameter sets in Table C1. Recall was calculated for three lead times: $t_{+1}$, $t_{+4}$ and $t_{+7}$. Recall can range from 0 (wrong) to 1 (perfect).

| Model no. | Recall | | | | | | | | |
| --- | --- | --- | --- | --- | --- | --- | --- | --- | --- |
| | Run 1 | | | Run 2 | | | Run 3 | | |
| | $t_{+1}$ | $t_{+4}$ | $t_{+7}$ | $t_{+1}$ | $t_{+4}$ | $t_{+7}$ | $t_{+1}$ | $t_{+4}$ | $t_{+7}$ |
| 1 | 0.89 | 0.88 | 0.85 | 0.73 | 0.70 | 0.70 | 0.80 | 0.80 | 0.80 |
| 2 | 0.83 | 0.85 | 0.85 | 0.75 | 0.88 | 0.83 | 0.78 | 0.73 | 0.78 |
| 3 | 0.88 | 0.88 | 0.85 | 0.68 | 0.75 | 0.75 | 0.75 | 0.75 | 0.78 |
| 4 | 0.80 | 0.83 | 0.85 | 0.78 | 0.78 | 0.83 | 0.85 | 0.85 | 0.83 |
| 5 | 0.80 | 0.80 | 0.85 | 0.88 | 0.90 | 0.90 | 0.73 | 0.68 | 0.70 |
| 6 | 0.75 | 0.73 | 0.75 | 0.83 | 0.83 | 0.80 | 0.80 | 0.78 | 0.78 |
| 7 | 0.70 | 0.73 | 0.65 | 0.75 | 0.73 | 0.73 | 0.75 | 0.78 | 0.83 |
| 8 | 0.83 | 0.83 | 0.80 | 0.83 | 0.75 | 0.78 | 0.80 | 0.83 | 0.85 |
| 9 | 0.75 | 0.83 | 0.83 | 0.78 | 0.83 | 0.83 | 0.80 | 0.73 | 0.80 |
| 10 | 0.88 | 0.88 | 0.88 | 0.78 | 0.73 | 0.75 | 0.80 | 0.85 | 0.83 |
| 11 | 0.80 | 0.83 | 0.83 | 0.83 | 0.78 | 0.80 | 0.88 | 0.85 | 0.85 |
| 12 | 0.85 | 0.85 | 0.83 | 0.80 | 0.78 | 0.78 | 0.73 | 0.75 | 0.78 |
| 13 | 0.73 | 0.70 | 0.65 | 0.73 | 0.70 | 0.68 | 0.78 | 0.83 | 0.83 |
| 14 | 0.83 | 0.83 | 0.80 | 0.70 | 0.70 | 0.70 | 0.88 | 0.88 | 0.88 |
| 15 | 0.80 | 0.85 | 0.83 | 0.90 | 0.95 | 0.93 | 0.83 | 0.88 | 0.88 |
| 16 | 0.88 | 0.88 | 0.88 | 0.75 | 0.78 | 0.73 | 0.78 | 0.88 | 0.85 |
| 17 | 0.78 | 0.78 | 0.80 | 0.75 | 0.73 | 0.73 | 0.78 | 0.80 | 0.78 |
| 18 | 0.73 | 0.80 | 0.78 | 0.80 | 0.83 | 0.83 | 0.83 | 0.83 | 0.83 |
| 19 | 0.75 | 0.78 | 0.75 | 0.78 | 0.83 | 0.80 | 0.83 | 0.83 | 0.83 |
| 20 | 0.80 | 0.80 | 0.85 | 0.80 | 0.78 | 0.80 | 0.80 | 0.80 | 0.83 |
| *21* | *0.78* | *0.78* | *0.78* | *0.83* | *0.90* | *0.90* | *0.83* | *0.83* | *0.88* |
| 22 | 0.75 | 0.75 | 0.75 | 0.85 | 0.83 | 0.80 | 0.83 | 0.85 | 0.93 |
| 23 | 0.85 | 0.85 | 0.90 | 0.83 | 0.88 | 0.88 | 0.75 | 0.78 | 0.78 |
| 24 | 0.80 | 0.83 | 0.83 | 0.75 | 0.70 | 0.73 | 0.75 | 0.80 | 0.80 |
| 25 | 0.80 | 0.68 | 0.68 | 0.75 | 0.83 | 0.80 | 0.80 | 0.85 | 0.85 |
| 26 | 0.88 | 0.88 | 0.90 | 0.78 | 0.75 | 0.83 | 0.70 | 0.68 | 0.65 |
| 27 | 0.78 | 0.73 | 0.75 | 0.75 | 0.78 | 0.78 | 0.73 | 0.80 | 0.80 |
| 28 | 0.83 | 0.83 | 0.83 | 0.80 | 0.83 | 0.90 | 0.83 | 0.85 | 0.80 |
| 29 | 0.83 | 0.88 | 0.88 | 0.88 | 0.85 | 0.83 | 0.85 | 0.88 | 0.93 |
| 30 | 0.83 | 0.80 | 0.80 | 0.85 | 0.85 | 0.85 | 0.78 | 0.80 | 0.80 |
| 31 | 0.93 | 0.85 | 0.88 | 0.83 | 0.78 | 0.78 | 0.78 | 0.80 | 0.75 |
| 32 | 0.93 | 0.95 | 0.95 | 0.83 | 0.90 | 0.88 | 0.85 | 0.80 | 0.83 |
| 33 | 0.75 | 0.63 | 0.63 | 0.83 | 0.83 | 0.83 | 0.80 | 0.88 | 0.88 |
| 34 | 0.75 | 0.80 | 0.75 | 0.78 | 0.80 | 0.83 | 0.75 | 0.75 | 0.75 |
| 35 | 0.83 | 0.83 | 0.80 | 0.80 | 0.80 | 0.80 | 0.78 | 0.83 | 0.83 |
| 36 | 0.78 | 0.73 | 0.75 | 0.85 | 0.88 | 0.88 | 0.85 | 0.88 | 0.85 |
| 37 | 0.75 | 0.85 | 0.85 | 0.78 | 0.80 | 0.80 | 0.83 | 0.83 | 0.78 |

**Table C5.** F1-score for three randomly initialized models per hyperparameter setup. Every run shows the results of one model. The F1-score is the harmonic mean of precision and recall. Model numbers correspond to the hyperparameter sets in Table C1. F1-score was calculated for three lead times: $t_{+1}$, $t_{+4}$ and $t_{+7}$. F1 can range from 0 (wrong) to 1 (perfect).

| Model no. | F1-score | | | | | | | | |
| --- | --- | --- | --- | --- | --- | --- | --- | --- | --- |
| | Run 1 | | | Run 2 | | | Run 3 | | |
| | $t_{+1}$ | $t_{+4}$ | $t_{+7}$ | $t_{+1}$ | $t_{+4}$ | $t_{+7}$ | $t_{+1}$ | $t_{+4}$ | $t_{+7}$ |
| 1 | 0.86 | 0.89 | 0.86 | 0.81 | 0.81 | 0.81 | 0.81 | 0.81 | 0.81 |
| 2 | 0.85 | 0.83 | 0.85 | 0.79 | 0.85 | 0.83 | 0.81 | 0.81 | 0.84 |
| 3 | 0.88 | 0.84 | 0.82 | 0.79 | 0.85 | 0.83 | 0.80 | 0.81 | 0.83 |
| 4 | 0.83 | 0.83 | 0.80 | 0.84 | 0.85 | 0.86 | 0.91 | 0.86 | 0.86 |
| 5 | 0.84 | 0.83 | 0.83 | 0.84 | 0.84 | 0.80 | 0.82 | 0.79 | 0.80 |
| 6 | 0.85 | 0.79 | 0.75 | 0.85 | 0.77 | 0.69 | 0.83 | 0.82 | 0.80 |
| 7 | 0.80 | 0.83 | 0.78 | 0.83 | 0.81 | 0.81 | 0.79 | 0.82 | 0.83 |
| 8 | 0.84 | 0.81 | 0.79 | 0.88 | 0.85 | 0.85 | 0.86 | 0.80 | 0.81 |
| 9 | 0.80 | 0.86 | 0.81 | 0.84 | 0.86 | 0.78 | 0.85 | 0.80 | 0.84 |
| 10 | 0.82 | 0.80 | 0.78 | 0.83 | 0.81 | 0.81 | 0.81 | 0.86 | 0.82 |
| 11 | 0.84 | 0.85 | 0.81 | 0.84 | 0.83 | 0.83 | 0.90 | 0.84 | 0.83 |
| 12 | 0.84 | 0.84 | 0.84 | 0.84 | 0.83 | 0.80 | 0.81 | 0.82 | 0.84 |
| 13 | 0.83 | 0.82 | 0.79 | 0.82 | 0.81 | 0.80 | 0.82 | 0.86 | 0.84 |
| 14 | 0.84 | 0.84 | 0.79 | 0.80 | 0.81 | 0.82 | 0.84 | 0.83 | 0.83 |
| 15 | 0.82 | 0.88 | 0.87 | 0.87 | 0.89 | 0.85 | 0.86 | 0.84 | 0.80 |
| 16 | 0.88 | 0.84 | 0.85 | 0.80 | 0.83 | 0.81 | 0.82 | 0.85 | 0.83 |
| 17 | 0.80 | 0.78 | 0.81 | 0.83 | 0.81 | 0.82 | 0.83 | 0.83 | 0.81 |
| 18 | 0.79 | 0.84 | 0.84 | 0.80 | 0.85 | 0.80 | 0.86 | 0.84 | 0.86 |
| 19 | 0.83 | 0.86 | 0.83 | 0.84 | 0.87 | 0.86 | 0.85 | 0.85 | 0.85 |
| 20 | 0.86 | 0.80 | 0.76 | 0.83 | 0.84 | 0.84 | 0.86 | 0.85 | 0.86 |
| *21* | *0.84* | *0.83* | *0.84* | *0.85* | *0.86* | *0.85* | *0.87* | *0.87* | *0.86* |
| 22 | 0.82 | 0.82 | 0.82 | 0.88 | 0.86 | 0.84 | 0.85 | 0.84 | 0.85 |
| 23 | 0.91 | 0.84 | 0.85 | 0.81 | 0.85 | 0.84 | 0.85 | 0.86 | 0.84 |
| 24 | 0.82 | 0.86 | 0.85 | 0.83 | 0.81 | 0.82 | 0.81 | 0.83 | 0.83 |
| 25 | 0.84 | 0.77 | 0.76 | 0.80 | 0.86 | 0.81 | 0.85 | 0.86 | 0.86 |
| 26 | 0.85 | 0.79 | 0.73 | 0.85 | 0.85 | 0.88 | 0.82 | 0.79 | 0.77 |
| 27 | 0.85 | 0.82 | 0.82 | 0.82 | 0.84 | 0.82 | 0.80 | 0.84 | 0.83 |
| 28 | 0.85 | 0.86 | 0.84 | 0.83 | 0.87 | 0.86 | 0.87 | 0.88 | 0.85 |
| 29 | 0.86 | 0.91 | 0.89 | 0.89 | 0.84 | 0.83 | 0.86 | 0.84 | 0.85 |
| 30 | 0.83 | 0.83 | 0.82 | 0.85 | 0.85 | 0.85 | 0.83 | 0.85 | 0.85 |
| 31 | 0.95 | 0.87 | 0.88 | 0.88 | 0.85 | 0.84 | 0.84 | 0.88 | 0.82 |
| 32 | 0.89 | 0.90 | 0.84 | 0.87 | 0.88 | 0.85 | 0.91 | 0.89 | 0.91 |
| 33 | 0.85 | 0.76 | 0.76 | 0.89 | 0.89 | 0.89 | 0.83 | 0.88 | 0.85 |
| 34 | 0.80 | 0.83 | 0.80 | 0.82 | 0.83 | 0.83 | 0.82 | 0.82 | 0.81 |
| 35 | 0.85 | 0.85 | 0.83 | 0.81 | 0.81 | 0.82 | 0.83 | 0.87 | 0.85 |
| 36 | 0.84 | 0.83 | 0.85 | 0.87 | 0.85 | 0.85 | 0.87 | 0.90 | 0.88 |
| 37 | 0.82 | 0.87 | 0.86 | 0.85 | 0.84 | 0.77 | 0.84 | 0.86 | 0.83 |

Table C1 shows the combinations of hyperparameters that were evaluated as described in Sect. 2.3.5. We also show the results of testing in terms of RMSE (Table C2), precision (Table C3), recall (Table C4) and F1-score (harmonic mean of precision and recall) (Table C5). Based on the results of this tuning process, model 21 was chosen as the final setup (Table 2).

## Appendix D: Results ensemble members

Figures D1 & D2 show the forecasts and fits for the training dataset (2011–2017) and test dataset (2018–2020) as separate ensemble members.

*Author contributions.* AW designed the general goals as part of the SALTI Solutions research project, which were further refined by all authors. Data were obtained by BW, using a script designed by FB. BW designed the models and performed the data analysis with frequent consultation from the other authors. FB contributed to improvements of the models. BW wrote this paper and created the figures. CB, FB and AW reviewed the paper and figures. AW secured the necessary financial support for this project and handled its administration. CB and AW supervise BW in his PhD programme.

*Competing interests.* No competing interests are present.

*Acknowledgements.* This research was conducted as part of the SALTI Solutions project, which is made possible by the Netherlands Scientific Organization (NWO) (NWO perspectief programme 18-32). We thank our research partners and stakeholders in the SALTI Solutions project for their advice and feedback during the development of the research methodology and evaluation of the results. Julie Pietrzak, Jaap Kwadijk, Meinard Tiessen, Wouter Kranenburg, Jos Boontjes-Witkamp, Meinte Blaas and Vincent Beijk played an important role in this process. Thanks to Ymkje Huismans, Hans Korving, Klaas-Jan van Heeringen and Vincent Vuik for the information they provided about the study area. Thanks to Jan Verkade for providing instructions on probabilistic forecasting. The online machine learning courses of Andrew Ng and Sreenivas Bhattiprolu were very helpful for model design. We also thank several colleagues at the Hydrology and Quantitative Water Management group for proofreading parts of this manuscript. Finally, we are grateful for the detailed and constructive feedback of two anonymous reviewers.

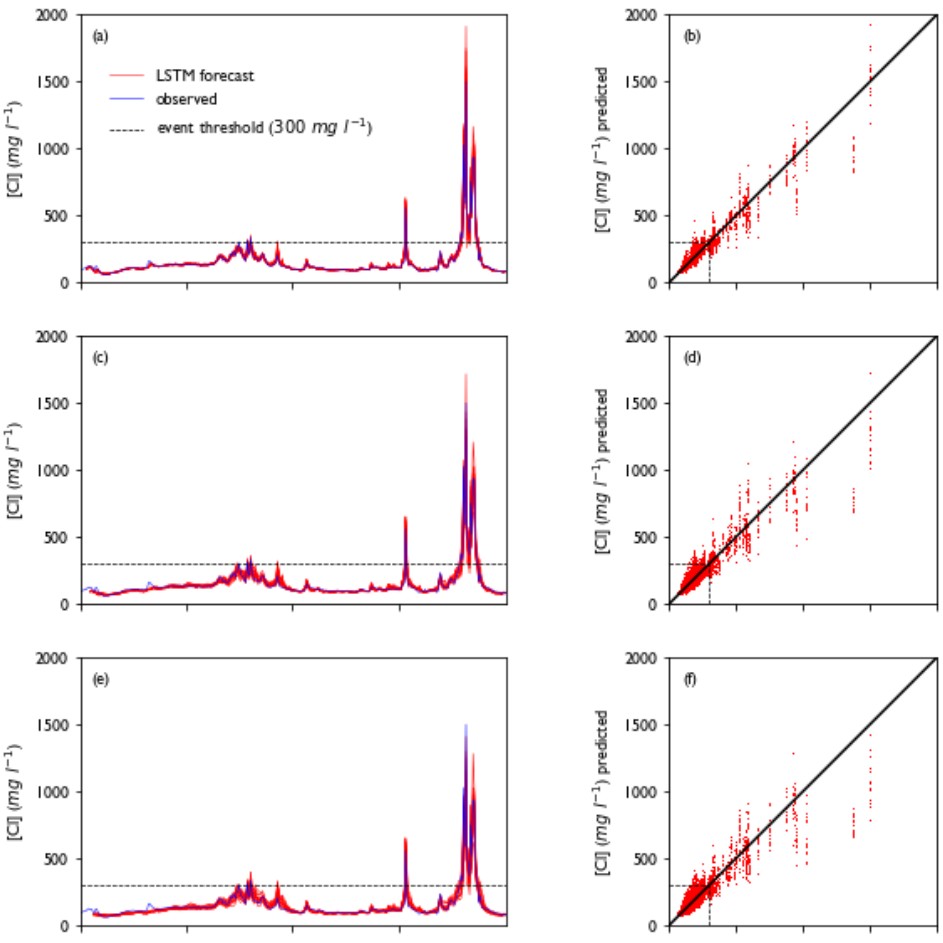

**Figure D1.** Model performance for the training period (2011–2017). Panels (a), (c) and (e) show a collection of forecasts of ClKr400Mean made with lead times of (a) 1 day, (c) 4 days and (e) 7 days for the year 2011, along with observed values. The predicted value is given as an ensemble prediction for each day of the year, created with the lead time mentioned. For each ensemble member, a separate line is plotted. Panels (b), (d) and (f) show predicted vs. observed values for the full training dataset with lead times of 1, 4 and 7 days, respectively. For each day in the training dataset, a single dot represents the prediction of a single ensemble member. This figure is summarized in Fig. 5.

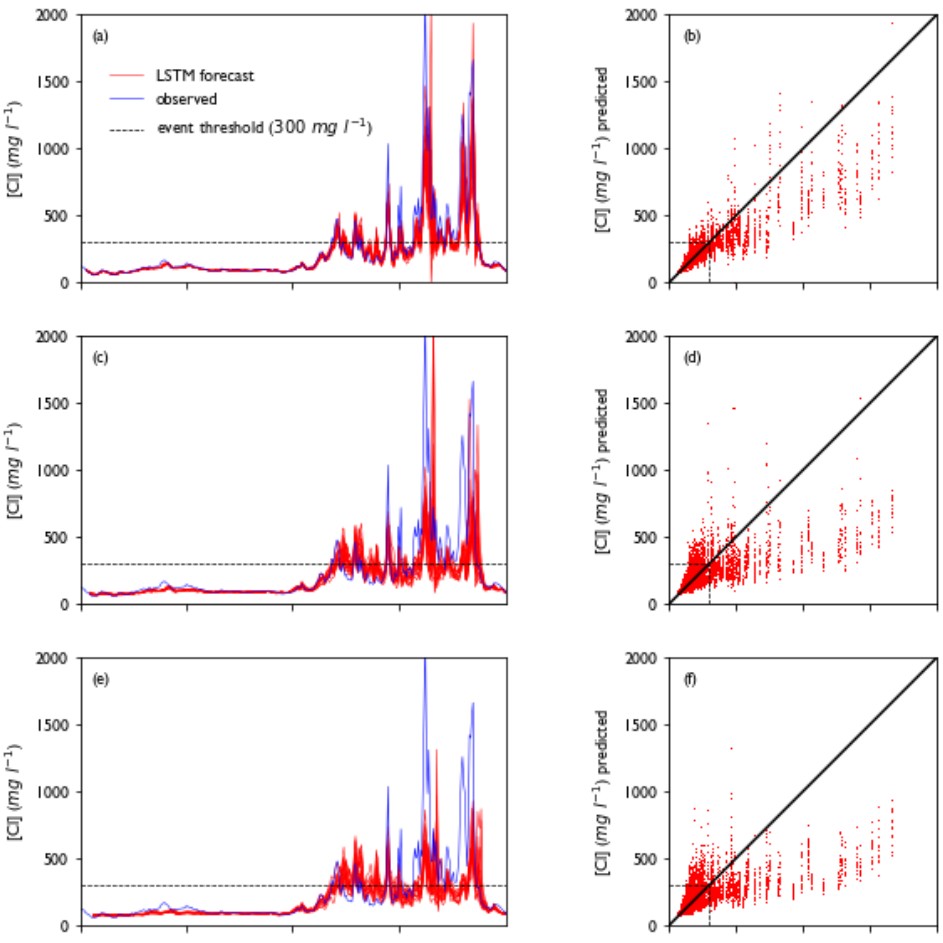

**Figure D2.** Model performance for the training period (2018–2020). Panels (a), (c) and (e) show a collection of forecasts of ClKr400Mean made with lead times of (a) 1 day, (c) 4 days and (e) 7 days for the year 2011, along with observed values. The predicted value is given as an ensemble prediction for each day of the year, created with the lead time mentioned. For each ensemble member, a separate line is plotted. Panels (b), (d) and (f) show predicted vs. observed values for the full training dataset with lead times of 1, 4 and 7 days, respectively. For each day in the training dataset, a single dot represents the prediction of a single ensemble member. This figure is summarized in Fig. 7.

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
