# Peer review of "Forecasting estuarine salt intrusion in the Rhine-Meuse delta using an LSTM model"

_EGUsphere, 2023_

## Author Response (AR1)

**Response to reviewer 1's comments**

The manuscript by Bas J.M. Wullems et al. presents a new modeling framework to simulate salt intrusion in the Rhine-Meuse delta, the Netherlands, up to 7 days ahead. The modeling framework, which is based on a LSTM model, is build and tested for one specific location based on input variables including chloride concentrations upstream and downstream locations, discharge, water levels and wind speed observation and evaluated using RMSE, precision and recall scores. Compared to process based models, the proposed machine learning framework has computational benefits and the ability to handle large amounts of data in an efficient manner. The authors state that these type of models would be able to support local water management and that a similar framework could work in other deltas as well, if of similar structure.

While this study is overall well written, interesting and introduces a new modelling framework to simulate salt intrusion in delta regions based on machine learning, some changes regarding structure and revisiting some of the sections for more clarification is proposed. These suggestions will be listed under major comments, while the minor comments cover mainly points on consistency, some clarifications and figure improvement suggestions.

*Thank you very much for taking the time to read this manuscript. We appreciate the extensive and detailed comments and questions and the general constructive tone of the review. We have considered all your comments and followed most of your suggestions. In case we did not, we have explained why. Below, your comments are listed, with a reply below in italics.*

Major comments:

Sections that could use more clarification or additional information include:

Abstract: revisit abstract to include additional information which is currently missing to make the abstract more clear and informative. E.g. include info from conclusions (line 341- 344) on model details as well, as well as last two sentences.

*Thank you for this suggestion. We added more information from the conclusion to the abstract.*

Introduction:

From line 48 onward the idea of the study and potential successful implementation are listed. I would however add a line with the clear goal and ambition of the study as well before that.

*We added a line to clarify the main goal of the study.*

line 55: 1) suggests that the study will also include an overview on how the suitable forecasting location will be found, however the location is introduced in line 79 without further explanation. The reasoning or exploration for this location could be included in the study area or statement 1) could be removed.

*That is a very good point. We added a sentence to clarify the choice for Krimpen aan den IJssel.*

To make the link between introduction and discussion, the statements in line 55-57 could briefly be revisited in the discussion and additional examples of other salt intrusion prone regions and modelling (assuming line 24-30 is focusing on the Rhine-Meuse delta) could be added to put the current study in a bigger picture.

*We added several sentences to the beginning of the discussion to link back to the end of the introduction. We also added some examples from other areas in the introduction and will give further attention to those in the discussion.*

Methods:

line 169-172: 'The model is trained to predict chloride concentrations on t+1, which is the first forecast. The forecast is then added to the record of chloride concentrations and used to forecast the next timestep…" Does this mean that for a chloride concentration forecast for t+7 , the simulation is based on the simulated concentrations of the previous timesteps? Is each simulated timestep evaluated separately before? Does this procedure not introduce additional uncertainty compared to if only observations are used to train?  Or are observations added for the specific timesteps during training?

*The model is only trained to forecast one step ahead. The simulation, however, is performed up to seven steps ahead, using the forecasts of the previous timestep. It is true that using simulated rather than observed values introduces extra uncertainty, but that is inherent to multi-step forecasting. To use only observations, we would have to train separate models for each lead time. We chose not to do this, as this might lead to models that suddenly shift variable weights from one day to the next, making it even harder to link the model behaviour to physical processes. Instead, the LSTM model can partially account for that by gradually changing variable weights over time. As the model is only trained on $t_{+1}$, the later predictions do not affect the outcome of the training process. They were however used during hyperparameter tuning and can be seen as an extra validation step.*

Line 194: how was the range of the hyperparameters to be tested defined?

*The setup of model no. 1 stays as close as possible to the LSTM setups found in manuals for neural network design (e.g. https://github.com/bnsreenu/python_for_microscopists/blob/master/166a-Intro_to_time_series_Forecasting_using_LSTM.py and https://towardsdatascience.com/multivariate-timeseries-forecast-with-lead-and-lag-timesteps-using-lstm-1a34915f08a ). From there we proceeded by adjusting each parameter in steps of a factor 2 for layer sizes or 0.1 for the dropout ratios. We kept adjusting one parameter at a time until we found a marked decrease in performance. We then recombined the best performing setups and searched for an optimum.*

line 212-214: for the sensitivity analysis the authors mention that they perturbed the test data set by adding 0.2 (in normalized units). Was this done for every input variable at the same time? Or for every variable separately and the model rerun every time? The manuscripts suggests the first, whereas I would expect the second option to give a more detailed sensitivity analysis. What would be the expected outcome if one would do the sensitivity analysis by changing only one variable at a time?

*Thank you for pointing this out. The sensitivity analysis is performed by perturbing a single variable at a time and then rerunning the model ensemble. Perturbing all variables at once would make the results of the analysis much harder to interpret. Since the text now seems to suggest that is what we did, we have rewritten this paragraph to correctly describe our actions.*

Proposing change of manuscript structure regarding methods and results:

Study area and Methods: propose to combine these into one 'material and methods', where study area could be moved directly, following sections would be data, model, sensitivity analysis.  data would include collection and timeseries exploration, model  would include model architecture, feature selection, evaluation/performance metrics, model tuning and testing,  and sensitivity analysis searate – so mostly moving sections around.

*We followed your advice and created a material and methods section.*

Potentially Fig 3 could be extended to a larger overview of all the steps included to build the model framework, not just the model architecture (also see more detailed comment below in minor comments section), to give the reader a full overview of the whole process

*We created a figure detailing the workflow and linking each step to a subsection in the manuscript.*

Results: Adding some of the observations and interpretation listed in the first discussion paragraph to the described results would support the reader and create a better understanding immediately

**We added some of the more direct interpretations to the results section, but kept most of it in the discussion section.**

Discussion: if possible also include comparison to other salinity studies, some are mentioned in the introduction but would be nice to come back to them and compare your work and put it in a bigger picture

*We added comparisons to studies of the Changjiang, Yangtze, Sumjin, Merrimack, La Comté and Pearl rivers.*

Minor comments:

- Title: a LSTM model

*Since the acronym is pronounced with a vowel sound (El-Es-Tee-Em),* an *is the correct article here.*

- Abstract:

    - line 4: in the Rhine-Meuse delta, the Netherlands.

    *Added*

- Methods:

    - line 65: Nederrijn/Lek or Lek/Nederrijn? if there's no specific reason to switch up order, try and keep consistency throughout manuscript

    *Changed to Nederrijn/Lek throughout the manuscript*

    - line 80: spacing too much after (Sect. 3.6).

    *removed*

    - Fig 1: like this overview figure idea! maybe some things to consider: make filled areas more transparent, station names of measurement locations

larger, model location bold, red dot to make it easier for the reader to find it, are city names relevant?

*We simplified the overview figure so that we could zoom in more on the study area and no longer need the double map. We also made the symbols simpler, to make the map less busy. We kept the names of major cities in, so that the reader can interpret the importance of waterways for residents.*

- Fig 1: caption what are the abbreviations PDOK and KNMI standing for?

*Translated names added to the reference list.*

- line 86: consider adding the information on data availability and links from Rijkswaterstaat KNMI and PDOK int the Code and data availability section at the end of the manuscript

*Added links to Rijkswaterstaat and KNMI data portals. PDOK wasn't added, as it was only used for mapping and therefore its data don't directly relate to the results.*

- line 90: are the stage-discharge relationships used in the end? Do these stations not have water level observations as well?

*Stage-discharge relationships are used by Rijkswaterstaat to calculate the discharges they end up publishing. You are right that we could use water levels as well. This might reduce uncertainty. On the other hand, calculations relating to water distribution and upstream supply of water centre around discharge, so for forecasting and management these may be more practical. We will take this into consideration for the larger-scale forecasting system.*

- line 103/104: maybe you could include a line for the threshold in Fig 2 to highlight salt intrusion events

*Good suggestion, we added this line.*

- line 105: '...increased water levels during storms (e.g. Fig. 2(c,e)) – wind speed doesn't really show, maybe remove e) in this reference

*Storms might have been an overstatement. We have rephrased this to 'periods with relatively high wind speeds', to indicate the fact that in these situations, daily mean water levels rise in concert with wind speeds.*

- line 109: Appendix A

*Done*

- Fig 2: a) legend indicates two levels, however only -5.5 in orange is shown. Missing -4.0 or the same values as for -5.5?

*The values are very close together, so the lines are hard to see separately in this figure.*

- Fig 2: caption need more explanation to guide the reader through them. what's the key message these figs convey?

*We extended the caption to describe the contents of each panel, paying special attention to the difference between the chloride peaks in January and other months.*

- Fig 2: compared to following figures indication of a), b), c) are given as titles, while following figures have the annotation within the subfigures – chose one option for consistency

*We have changed the titles to annotations within the subfigures and but the description in the caption.*

- Table 1: consider removing column t-7...t-5 and just mention it was tested (currently not even mentioned)

*We added information explaining why we dropped t-7...t-5 have added Boruta results to the appendix. We prefer to keep the empty column in, as it clarifies how we removed surplus variables and timesteps from the analysis.*

- line 125: refer back to Fig 1 with the map for the locations

*Reference added*

- line 139: wind, change to wind speed to keep consistency

*Done*

- line 140: another model? referring to a specific one or generally?

*Generally. This describes how we imagine our model being implemented in a forecasting system which uses a range of meteorological, hydrological and*

*hydraulic models, which can be run in series or in parallel. In the current setup, our model can only be run successfully after a hydraulic model has been run to predict water levels at $t_{+1}$. The same applies to discharge and wind speed. Since prediction of these variables is already an integrated part of operational water management, we do not expect this to be a problem.*

- line 141: maybe remove the omitted features in Table 1 and this line?

*We rewrote the line to be more general, but kept the omitted variables in the table, to make sure it is clear which variables were initially considered but dropped in the end.*

- line 144: future chloride concentrations – define future, is it near-future (upcoming days) or subseasonal...

*Up to 7 days ahead, we added this to the sentence.*

- Fig 3: rephrase salt to chloride for consistency

*Done*

- Fig 3: having the schematic overview of the machine learning model is useful. However, I would even extend it to include all steps before (data preparation, feature selection, preprocessing, etc.) and after (evaluation, sensitivity analysis) into that figure as well. Kind of make it a comprehensive overview which also guides throughout the subchapter in Chapter 3

*This is a very good suggestion. We chose to create an extra figure that outlines the whole data processing and forecasting procedure, which we. The current figure is retained to show the structure of the model itself.*

- line 151: 'preprocessing' – what steps were included in preprocessing? Some additional lines would be useful

*Preprocessing is limited to feature selection, interpolation and normalization. Feature selection  is mentioned in the previous section. We forgot to mention interpolation here, so we have added that to this paragraph.*

- line 152: rephrase: 'Measurements of chloride concentration for t-4 up to t0, as well as measurements of discharge, water level and wind speed for t-4 up to t+1 were used as input

*Done*

- line 150: double 'for documentation, see'

*Removed*

- line 160: 'quantity variables': be consistent in text and figure (change quantity data to quantity variables)

*Changed to quantity variables*

- line 160: salt timeseries – change to chloride timeseries (for continuity throughout manuscript)

*Changed to chloride timeseries*

- line 174 Root Mean Square Error (RMSE), introduce abbreviation first time and afterwards stick to using abbreviation

*Done*

- line 206 missing . after Appendix B

*Inserted*

- Results: check that all figures use same font family

*All figures now use the Gill Sans MT font.*

- line 226: missing bracket after Fig 4 (b,d,f)

*Inserted*

- Fig 4 and Fig 6, would it be possible to show the results of all ensemble members for a), c) and e) in addition to the ensemble prediction (median and range)? Doesn't need to be in this figure but would be nice for the Appendix

*We have added a figure with separate ensemble members to the appendix. We have also adapted panels b, d, and f in the main manuscript to show only a range. The heavy and busy figure with separate dots per ensemble member has also been moved to the appendix.*

- Fig 4 and Fig 6, would remove figure title as the info is included in the figure caption

*Done*

- Fig 4, curious whether it is known why/when the highest observation values on the right side of c), d), f) were observed? Outlier or specific event?

*The highest values correspond to the peak on 28 November 2011, which is also visible in panels a), c), e).*
- Fig 5, (a) Root Mean Squared Error, also consider changing all to RMSE after you introduced the abbreviation

*Changed to RMSE*

- line 244: less strong, find synonym (e.g. minor, little, smaller, etc.)

*Changed to smaller*

- line 245: wind speed seems less important, was it considered to be left out at some point? To reduce input variables for example

*We did consider this, but the Boruta analysis indicated that wind speed was moderately relevant, i.e. model performance was less good when it was left out. It is difficult to reconcile this with the small importance of this variable indicated by the sensitivity analysis. It might be possible that wind speed only plays a role in a certain regime, where its indirect influence through water level offers insufficient explanation, but we have not been able to find the exact situations where this occurs.*

- Fig 6, in c) and e) it looks a bit like there's a shift in the simulated peaks (especially Aug and end of the year (Nov/Dec), any idea why? Just curious

*It is hard to say for certain, but these shifts seem to occur especially when two peaks occur shortly after one another. What we think happens is that the rising limb from the previous peak is simulated to continue for longer than it does in practice. When a change in discharge or water level occurs, the response from the previous peak is then projected onto the current peak, simulating it to be as high as the previous one, or even higher. This would be in line with the sensitivity of predicted chloride concentrations to concentrations on the preceding day. This might also explain why some of these shifts seems stronger for the 4-day forecasts than for the 7-day forecasts: the added observations of discharge, water level and wind have a relatively larger influence for longer lead times, so that the exaggerated effect of repeating peaks is dampened somewhat.*

- Fig 6, would it be possible to change the color of the dots for 2018 in subfigures b), d) and f)? while a), c) and e) is showing only 2018, the others are covering 2018-2020

*Good idea, we also applied this to the 2011-2017 figure.*

- Fig 7, (a) Root Mean Squared Error, also consider changing all to RMSE after you introduced the abbreviation,

*Changed to RMSE*

- Fig 7, lead time (days) (as in Fig 5)

*Adjusted*

- Fig 7, so is your model performing better than the reference or not? try to give key message also in fig caption

*Thank you for this point of attention, we have adjusted the captions of both figure 5 and 7 (now figure 6 and 8).*

- Fig 8, smaller font size for x and y axis if possible, currently looks a bit out of balance.

*Done*

- Fig 8, The caption includes all the necessary information for a standalone interpretation, nice!

*Thank you!*

- Discussion:

- if interpretation of results is combined with results section the additional title limitation and outlook could be removed

*Thank you for your suggestion. We have chosen to keep the interpretation section in the discussion, because we like to have a part linking the various results together.*

- line 259: "… as water managers usually prefer to err on the side of caution.' Remove err or word missing?

*We rephrased this sentence to:* This is probably not desirable, as the consequences of a missed event are typically more problematic than the consequences of a false alarm

- line 270: Nederrijn/Lek or Lek/Nederrijn? if there's no specific reason to switch up order, try and keep consistency throughout manuscript

*Changed to Nederrijn/Lek throughout the manuscript*

- line 274: has the setup been tried without including wind speed? Maybe limiting the input variables even more would also lead to similar results? Just curious

*We tried leaving out wind speed and it did yield acceptable results. However, based on the Boruta analysis we could not justify leaving it out of the setup completely. We aim to get a better idea of the contribution of wind by comparing the model ensemble created for Krimpen aan den IJssel to other locations.*

- Conclusion:

  - line 347: salt concentration, change to chloride concentration to keep consistency throughout manuscript

  *Done*

- Appendix A: maybe add a short line on what is key message of Fig A1 (or add that to the figure caption)

  *We added some information on the interpretation of Fig A1.*

- Appendix B: tables following should have reference with table B1, B2, etc. currently still referencing Appendix A

  *Fixed*

- Table A1: how were the hyperparameters range to be tried chosen?

  *See reply to your comment on line 194.*

- Table A2, A3, A4 and Table A.5:

  - are Run1, Run2, Run3 the three different initialized models? Maybe dd the clarification to the table caption or adapt the table headers.

*Every run shows the results of one model. We added this information to the caption.*

- maybe also highlight the  model you used in the end (e.g. whole row in bold for model 21)

*Done*

- add reference to what the ideal score would be in the captions

*Added*

**Response to reviewer 2's comments**

This manuscript proposes the use of a machine learning (ML) model to forecast salt intrusion for a specific location in the Rhine-Meuse delta. It is suggested that this forecast, up to 7 days ahead, could be valuable for local water managers who currently use observations and operational rules to control water flow in smaller channels as to ensure a good water quality. The authors propose a model architecture, optimize the various training and hyper-parameters and compare the model results with a 'baseline'-model that maintains the last observation as a forecast. The authors further address the various limitations of the work and make suggestions for improvements. It is concluded that the model can accurately predict the timing of peak events, but that absolute concentrations are consistently underestimated.

The paper is well-written and the results seem robust and useful. Below are several comments and concerns that I hope prove useful for the authors:

*We are very grateful for your questions, that have helped us clarify and rethink some of our methodological decisions. While we have included some of your suggestions in the current manuscript, we have also taken several of them into consideration for the follow-up step in our research, for which the current step is a proof of concept. In this reaction, we explain these decisions. We also value the comments of the figures, which we believe to be easier to read now thanks to your suggestions.*

(1) LSTM models are typically suited very well for identifying relevant information in time series. However, the authors chose to limit the input time series to a small number of preceding days. The authors need to describe better how the models are run: is every forecast made using only the preceding 5/6 time steps? Does this mean that the internal state variables of the LSTM models are reset to certain initial conditions, independent of for example the period during the year? Is no warm-up period used? In case so, consider testing longer input sequences during training and testing. Further, the authors could argue better why they only tested the LSTM model architecture, and none of the many other ML algorithms that exist these days.

*Thank you for pointing this out. Based on the timeseries exploration, we concluded that most of the relevant variables change on a timescale of days to a week. Furthermore, from the Boruta analysis, we concluded that five timesteps should be sufficient. We added an appendix describing the Boruta procedure more fully. It is true that the period during the year is not considered in this analysis. In follow-up research, we will try including variables that account for seasonal effects, like the Day of Year. We will also consider much longer sequences (months), perhaps as a pre-selection between normal and extreme situations. For this paper, however, we stick to the timescale of days to weeks that is suggested by the timeseries exploration and Boruta analysis. As for the*

*choice for LSTM, we started with a multivariate linear regression model and then set up a feedforward neural network and an LSTM network. We found we could get better results by tuning the LSTM model than by tuning the other two, which is why we chose to continue developing that model architecture. We have added this consideration to the 'model architecture' section.*

(2) Daily min, mean and max chloride concentrations are probably highly correlated, and perhaps the time series of chloride concentrations can be considered as constituted of a 'slower' component dependent on the upstream discharge and downstream salinity and water levels, and a daily tidal component. The authors claim that the number of input variables is limited as to reduce redundancy. However, they still include min, mean and max chloride concentrations. Consider training the model on only mean concentrations, and perhaps add a second model component to adjust for the maximum and minimum concentrations of the tidal cycle if these are relevant model outputs.

*This is an excellent suggestion that we will definitely include when developing the extended model for the Rhine-Meuse delta. We kept the daily min, mean and max in based on the Boruta analysis, but we may find that better results for the whole area can be achieved using a different set of variables. Seperating the timeseries into a slow and a quick component seems very promising in that respect, so we will implement that. Training a model to predict the mean and then a second model for min and max may reduce total runtime, so we will try that as well.*

(3) The authors mention several times that one possible application of the models is to use forecasts to better control the local hydraulic infrastructure to for example flush channels with fresh water. Furthermore, The authors state (e.g. L52) that the studied estuary is intensively managed. However, from the text it is not clear whether those management operations are explicitly included in the inputs of the LSTM model. If not, this means that no scenario analyses can be conducted using the models (e.g. as mentioned on L279).

*At this point, management operations are not explicitly included in the model. This indeed limits the possibilities for scenario analysis. Hauswirth et al. (2021) also mention the difficulty of including management in a machine learning model for floods. Some scenario analysis is possible though, if adjusted discharges and water levels are used as inputs. We have added this clarification to the discussion in the same paragraph as the claim you mention.*

(4) Figures:

L84: Figure 1 is quite chaotic, and is inadequate of presenting the readers with a clear understanding of the study area and river network. Names of key localities or monitoring locations and rivers are not clear and sometimes even unreadable, especially in the centre of the bottom halve figure. I suggest combining a map of the

area with a schematized illustration of the river network and the interconnected river branches + locations of points of interest. Perhaps also indicate the locations of weirs and gates that are used to control the river network and flush salt water when concentrations are rising, as this is proposed as one of the major goals of the model.

*Thank you for your suggestions. We have taken them to heart and made a new overview map of the study area, that is calmer and easier to read. We added an inset map to show the location of the Rhine and Netherlands within Europe, so that we could zoom in more on the study area and no longer needed the double map. Weirs and gates will be a subject of a later step in our research, when we will give more attention to their exact location.*

L114: Figure 2: the events described in the text are not clearly identifiable from the graphs. Consider using a scatter plot to better compare relations between the time series that are described as the various events. Alternatively, grid lines or indications with arrows/symbols of the mentioned events, or the use of detailed plots of individual events discussed in the text could better support the claims.

*We have created zoomed in plots on the periods with events, to make the coinciding peaks more visible.*

(5) L98: splitting the time series in days with concentrations exceeding the threshold seems artificial, for example, some multi-day events of high concentrations could exist. It might be better to use for example a 'peak-over-threshold' analysis to properly select the number independent events in the training and testing data.

*It is true that several multi-day events occur, especially in 2018. However, predicting whether the elevated chloride concentrations will persist is also part of the forecasting objective. We therefore consider every single day as a separate event or non-event. In the further development of the forecasting system, we will consider using a more sophisticated method, considering the inception and duration of an event separately.*

(6) If the model only predicts the chloride concentration at 'Krimpen aan den IJsel', how is the concentration at Lekhaven predicted in forecasts?

*The model predicts concentrations at both locations, but is preferentially tuned to get a good prediction for Krimpen aan den IJssel.*

(7) L329: consider combining the comment of using ML models as emulators with the remark of L291: the use of detailed, physics-based models to generate training data that covers more extreme situations.

*We have moved these paragraphs closer together, so that this connection was easier to make.*

(8) The authors should better argue why the sensitivity analysis was conducted in this manner, by increasing the various inputs with 0.2 in normalized units.

*This is a good point. We started with a larger increase, but found this to correspond to combinations of water levels and discharges that were unlikely to occur. We also had some problems with the backtransformation of values that lead to negative chloride concentrations. We therefore decreased the stepsize to 0.2. We have added this explanation to the manuscript.*

(9) Language:

- Replace 'underpredicted' with 'underestimated' or another synonym.

  *Replaced with 'underestimated'*

- L150: typo: two times 'for documentation, see'...

  *fixed*

- L244: less strong: consider finding a synonym.

  *Replaced with 'small'*

- L301: scrap 'a' (last word of sentence).

  *Scrapped*